# ZBP1 promotes LPS-induced cell death and IL-1β release via RHIM-mediated interactions with RIPK1

Hayley I. Muendlein[1,5], Wilson M. Connolly[1,5], Zoie Magri [1], Irina Smirnova[1], Vladimir Ilyukha [2], Avishekh Gautam[3], Alexei Degterev[4] & Alexander Poltorak [1,2✉]

Inflammation and cell death are closely linked arms of the host immune response to infection, which when carefully balanced ensure host survival. One example of this balance is the tightly regulated transition from TNFR1-associated pro-inflammatory complex I to pro-death complex II. By contrast, here we show that a TRIF-dependent complex containing FADD, RIPK1 and caspase-8 (that we have termed the TRIFosome) mediates cell death in response to *Yersinia pseudotuberculosis* and LPS. Furthermore, we show that constitutive binding between ZBP1 and RIPK1 is essential for the initiation of TRIFosome interactions, caspase-8-mediated cell death and inflammasome activation, thus positioning ZBP1 as an effector of cell death in the context of bacterial blockade of pro-inflammatory signaling. Additionally, our findings offer an alternative to the TNFR1-dependent model of complex II assembly, by demonstrating pro-death complex formation reliant on TRIF signaling.

[1] Department of Immunology, Tufts University School of Medicine, Boston, MA 02111, USA. [2] Petrozavodsk State Ulniversity, Petrozavodsk, Republic of Karelia 185910, Russia. [3] Blood Cell Development and Function Program, Fox Chase Cancer Center, Philadelphia, PA 19111, USA. [4] Department of Developmental, Molecular and Chemical Biology, Tufts University School of Medicine, Boston, MA 02111, USA. [5]These authors contributed equally: Hayley I. Muendlein, Wilson M. Connolly. ✉email: Alexander.poltorak@tufts.edu

nnate immune cells rely on NF-κB and mitogen-activated protein kinase (MAPK) signaling cascades downstream of Toll-like receptors (TLRs) to protect the host against pathogens[1]. In response, successful pathogens employ mechanisms to circumvent these host inflammatory responses. For example, *Yersinia* species bacteria release effector proteins known as *Yersinia* outer proteins (Yops), which are capable of modulating these host responses in favor of bacterial survival and replication[2]. One such effector, YopJ blocks activation of the level 3 MAP kinase TAK1 (TGFβ-activated kinase), attenuating host inflammatory and pro-survival signaling[3,4]. In response to this blockade, host cells initiate rapid cell death that is crucial for successful bacterial clearance and host survival[5,6]. This positions cell death as an important host mechanism to counteract the inhibition of critical inflammatory signaling by pathogens.

Pyroptosis is an inflammatory form of cell death that occurs in response to pathogenic infection and is typically driven by caspase-1 (CASP1) and/or caspase-11 (CASP11)-mediated cleavage of gasdermin D (GSDMD), which drives death by generating membrane pores[7,8]. The inflammatory nature of pyroptosis is revealed via the inflammasome-dependent maturation and release of inflammatory IL-1β and IL-18 cytokines. IL-1β maturation often occurs downstream of the NLRP3 inflammasome that includes apoptosis-associated speck-like protein (ASC) and CASP1, which cleaves pro-IL-1β, enabling release of mature IL-1β via GSDMD pores[9,10].

Recently, we and others have identified a form of pyroptosis-like cell death that occurs in murine macrophages in response to *Yersinia* infection[11,12]. This YopJ-dependent death relied on the kinase activity of receptor-interacting serine/threonine-protein kinase 1 (RIPK1) to drive caspase-8 (CASP8) activation, and the cleavage of downstream caspases-1, 3, 7, 9, and 11[11,12]. Importantly, *Yersinia* infection induced cleavage of the pyroptosis effector GSDMD by CASP8[11,12]. However, CASP3 and CASP7-mediated activation of apoptotic effectors downstream of CASP8 also contributes to overall cell death in response to *Yersinia* infection[11], leading to a mixed cell death phenotype with contributions from both pyroptotic and apoptotic pathways. Similar signaling appears to drive TNF-induced cell death in the context of TAK1 inhibition[13], but the mechanisms underlying CASP8-mediated cell death have yet to be fully elucidated.

Here, we show that CASP8 activation occurs within a complex that is similar to the extensively characterized TNFR1-associated complex II, but is largely independent of TNF receptor 1 (TNFR1). Instead, the formation of this complex, which contains FADD, RIPK1, CASP8, and Z-DNA binding protein 1 (ZBP1) is dependent on TIR-domain-containing adapter-inducing interferon-β (TRIF), providing us with new insights into the regulation of pro-death signaling downstream of TRIF. In addition, we show that ZBP1, constitutively bound to RIPK1 via RHIM-mediated interactions facilitates interactions between CASP8, FADD, and RIPK1, promoting caspase-8-mediated cell death and inflammasome activation. To reflect the mandatory requirement for TRIF rather than for TNFR1 for these pro-death interactions involving ZBP1, we have named this complex the TRIFosome.

## Results

### TRIF is required for CASP8-mediated cell death

To aid in a thorough mechanistic inquiry of CASP8-mediated cell death, we used bone marrow-derived macrophages (BMDMs) activated with lipopolysaccharide (LPS) and the small-molecule inhibitor of TAK1, 5Z-7-Oxozeaenol (5z7) to mimic *Yersinia* infection[11,12]. As we have shown previously[11], LPS/5z7 induced a mixed cell death phenotype, that was partially dependent on the kinase activity of RIPK1, the apoptotic caspases CASP3 and CASP7, and

the effector of pyroptosis GSDMD, and entirely dependent on CASP8 (Fig. 1a). Using this model, we identified the TLR4 adaptor TRIF as an important mediator of CASP8-driven cell death, as death in *Trif*^−/− BMDMs treated with LPS/5z7 was abrogated to levels similar to those induced by 5z7 alone, which induces a mechanistically divergent, TNF-dependent form of necrotic cell death that is also dependent on CASP8 as described previously[11] (Fig. 1b). However, cell death occurred independently of the other main TLR4 adaptor MyD88 (Fig. 1b).

Although LPS/5z7 treatment induced rapid and abundant cleavage of caspase-8, 3, 7, 9, and GSDMD downstream of CASP8 in wild-type (B6) BMDMs, activation of these caspases was delayed and attenuated in the absence of TRIF (Fig. 1c). Cleavage of CASP9 is often indicative of mitochondrial or intrinsic apoptosis. Since it has recently been shown that GSDMD is capable of driving mitochondrial permeabilization, amplifying apoptotic and pyroptotic cell death[14], we were interested to see if GSDMD was required for CASP9 cleavage in our system. However, while CASP9 cleavage was dependent on CASP8, it occurred independently of GSDMD (Supplementary Fig. 1a, b). It is important to note that while deficiency in GSDMD did not affect caspase cleavage, membrane pore formation and cell death as measured by release of intracellular contents were abrogated in the absence of GSDMD (Supplementary Fig. 1b).

Reinforcing the importance of TRIF in initiating CASP8-mediated cell death pathways in response to LPS/5z7, LPS and TAK1-mediated phosphorylation of RIPK1 at S321 was attenuated in the absence of TRIF (Fig. 1d). In addition, phosphorylation of RIPK1 at S166, a hallmark of RIPK1 activation, as well as degradation of total RIPK1 were TRIF-dependent (Fig. 1d). To gain further insight into the mechanism of TRIF-driven pyroptosis, we compared the genes that were upregulated in response to LPS/5z7 in B6, *Myd88*^−/−, and *Trif*^−/− BMDMs (Fig. 1e). Of the 246 genes that were upregulated in susceptible (B6 and *Myd88*^−/−), but not protected *Trif*^−/− BMDMs, many were involved in Type I interferon responses (Fig. 1e, f, Supplementary Fig. 2a). This indicated that IFN induction downstream of TRIF might be responsible for regulating specific interferon-stimulated genes (ISGs) that drive CASP8-mediated cell death. Indeed, compared to B6, *Ifnb*^−/−, and *Ifnar*^−/− BMDMs exhibited decreased cell death, which could be enhanced by reconstitution with recombinant IFNβ prior to LPS/5z7 treatment in *Ifnb*^−/− macrophages (Supplementary Fig. 2b). Furthermore, inhibition of IFN signaling in B6 BMDMs using IFNAR blocking antibody was sufficient to recapitulate the decrease in death observed in *Ifnb*^−/− and *Ifnar*^−/− BMDMs (Supplementary Fig. 2b).

### ZBP1 promotes CASP8-mediated cell death and IL-1β release

Among many known ISGs differentially regulated between B6, *Myd88*^−/−, and *Trif*^−/− in response to LPS/5z7, Z-DNA binding protein 1 (ZBP1) was of particular interest (Supplementary Fig. 2a). ZBP1 is a RHIM (RIP homotypic interaction motif) domain-containing protein, which has been shown to mediate various forms of cell death via interactions with other RHIM domain-containing proteins including RIPK1 and RIPK3[15–21]. Since RHIM domain-containing proteins TRIF and RIPK1 have proven to play an important role in the regulation of CASP8-mediated cell death, it seemed likely that ZBP1 may also be involved (Fig. 2a). Confirming the classification of ZBP1 as an ISG in our system, ZBP1 levels were elevated in response to treatment with recombinant IFNβ (Supplementary Fig. 2c). In addition, ZBP1 mRNA levels were increased by stimulation with LPS and LPS/5z7 in a TRIF and IFNβ dependent manner, and inhibition of IFNAR signaling abrogated LPS-induced increases in ZBP1 protein levels (Supplementary Fig. 2d, e). Confirming

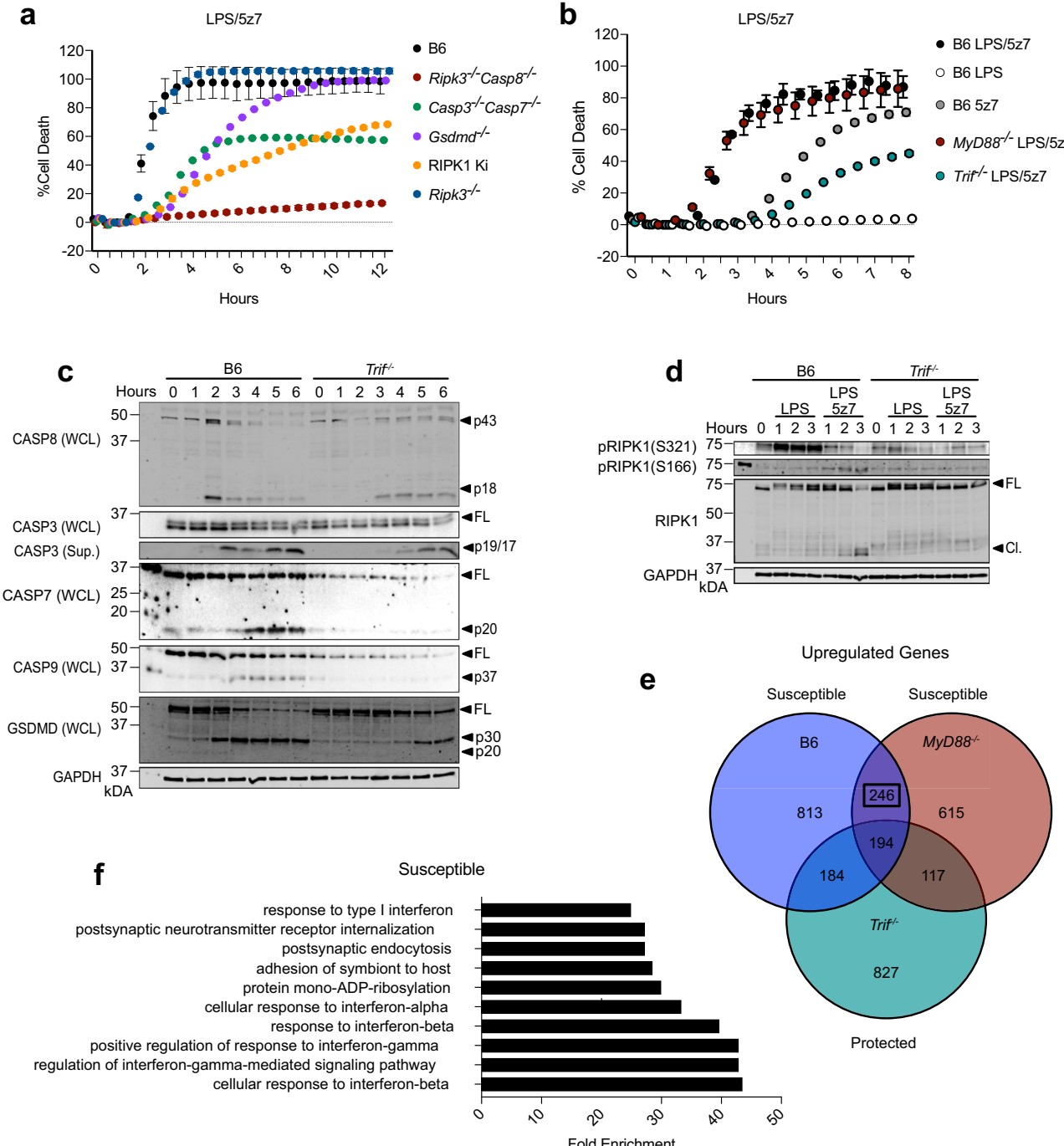

**Fig. 1 TRIF is required for CASP8-mediated cell death. a**, **b** Cell death over time as measured by propidium iodide incorporation in indicated BMDMs stimulated with LPS, 5z7, or LPS/5z7. **c** Full-length and cleaved products of indicated caspases and GSDMD from whole-cell lysates (WCL) or precipitated from the supernatant (Sup.) of B6 or $Trif^{-/-}$ BMDMs stimulated with LPS/5z7 for 1–6 h. **d** Levels of total, pRIPK1 (S166), and pRIPK1 (S321) in B6 or $Trif^{-/-}$ BMDMs stimulated with LPS or LPS/5z7 for 1–3 h. **e** Comparison of genes upregulated (>1.7-fold) after 1 h of LPS/5z7 treatment compared to unstimulated in B6, $Myd88^{-/-}$, or $Trif^{-/-}$. **f** Top upregulated pathways identified in B6 and $Myd88^{-/-}$ but not $Trif^{-/-}$ BMDMs after stimulation with LPS/5z7. Data from cell death assays and western blots are representative of 3 or more biologically independent experiments, cell death data are presented as the mean ± SD of triplicate wells, $n = 10,000$ cells examined in three individual wells. Source data for all experiments are provided as a Source data file. See also Supplementary Fig. 1.

our hypothesis that ZBP1 may play a role in the regulation of CASP8-mediated cell death, $Zbp1^{-/-}$ BMDMs exhibited attenuated death as measured by propidium iodide (PI) incorporation and lactate dehydrogenase (LDH) release in response to LPS/5z7, similar to what we observed in the absence of TRIF and the kinase activity of RIPK1 (Fig. 2b, c). Importantly, while ZBP1

deficiency attenuated cell death in response to LPS/5z7, $Zbp1^{-/-}$ BMDMs recapitulated B6 in response to 5z7 treatment alone, indicating that the role for ZBP1 was specific to LPS/5z7-induced cell death (Supplementary Fig. 2f). However, cell death was not completely abrogated in the absence of ZBP1, and like B6, the remaining cell death in $Zbp1^{-/-}$ relied on CASP8 and RIPK1

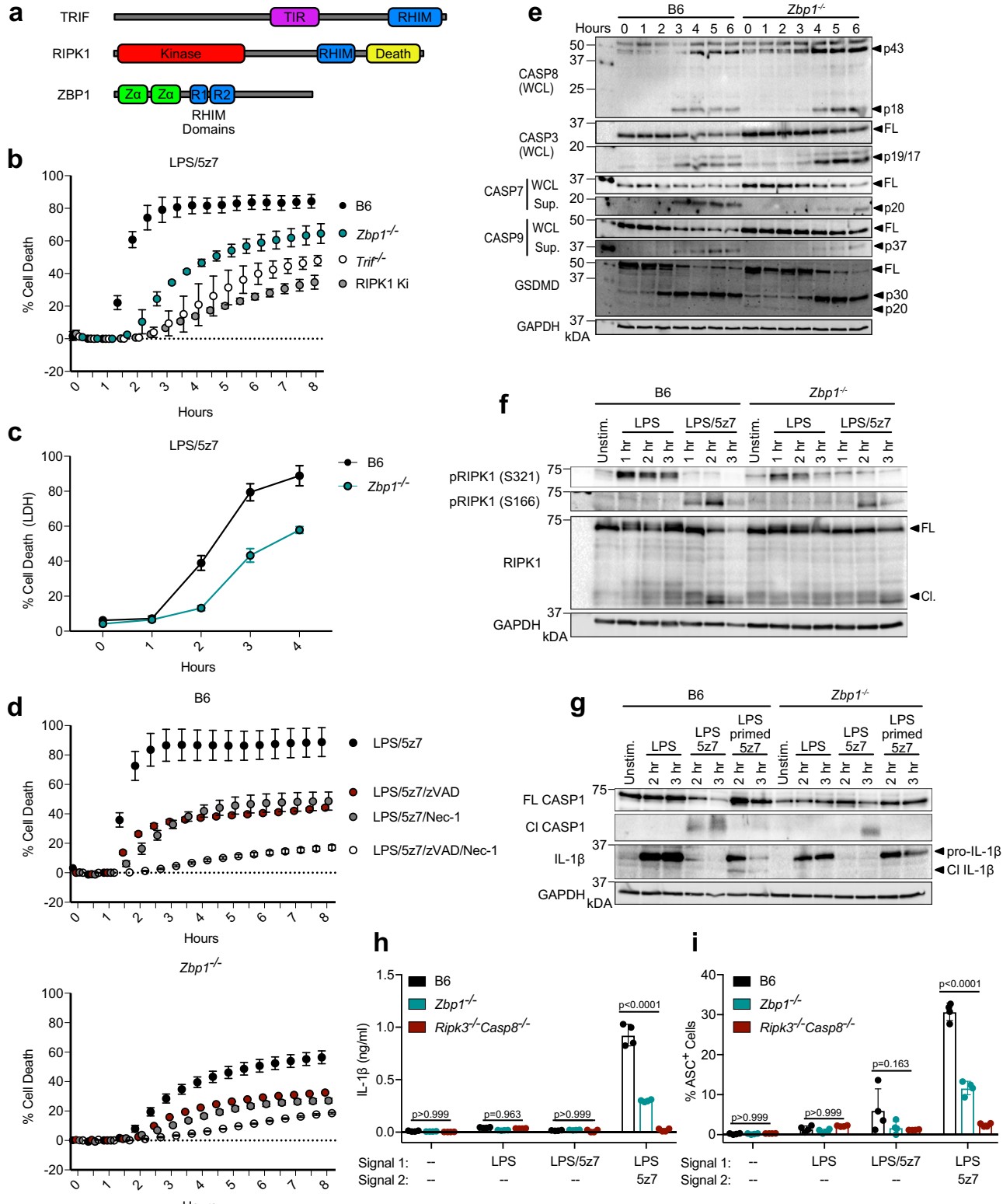

kinase activity, as treatment with the caspase inhibitor zVAD and/or the RIPK1 kinase inhibitor Nec-1 decreased death (Fig. 2d). This suggested that ZBP1 deficiency alters the kinetics and extent, but not the mode of cell death. While death was delayed in the absence of ZBP1, TRIF-deficiency provided stronger protection, supporting a role for ZBP1 downstream of TRIF-mediated signaling (Fig. 2b). In confirmation of the mechanism of cell death occurring in $Zbp1^{-/-}$ BMDMs, LPS/5z7

treatment resulted in cleavage of CASP8, 3, 7, 9, and GSDMD (Fig. 2e), indicating that ZBP1 is involved in the regulation of both apoptotic and pyroptotic cell death pathways in response to LPS/5z7. While this was in stark contrast to the complete abrogation of caspase and GSDMD activation observed in $Trif^{-/-}$, the kinetics of these cleavage events were delayed by ZBP1 deficiency (Figs. 1c, 2e). Furthermore, TAK1 and MAPK-mediated phosphorylation of RIPK1 at S321, and LPS/5z7-mediated

**Fig. 2 ZBP1 promotes CASP8-mediated cell death and IL-1β release. a** Schematic of protein domains in TRIF, RIPK1, and ZBP1. **b, c** Cell death over time in indicated BMDMs stimulated with LPS/5z7 as measured by (**b**) propidium iodide (PI) incorporation or (**c**) LDH release. **d** Cell death over time as measured by PI incorporation in B6 and *Zbp1*−/− BMDMs treated as indicated. **e** Full-length and cleaved products of indicated caspases and GSDMD from whole-cell lysates (WCL) or precipitated from the supernatant (Sup.) of B6 or *Zbp1*−/− BMDMs stimulated with LPS/5z7 for 1–6 h. **f** Levels of total, pRIPK1 (S166), and pRIPK1 (S321) in B6 or *Zbp1*−/− BMDMs stimulated with LPS or LPS/5z7 for 1–3 h. **g** Full-length and cleaved products of CASP1 and IL-1β from whole-cell lysates of B6 and *Zbp1*−/− BMDMs stimulated as indicated for 1–3 h. **h** IL-1β release after 6 h and (**i**) percentage of ASC+ cells after 2 h in B6, *Zbp1*−/−, and *Ripk3*−/−*Casp8*−/− BMDMs stimulated as indicated. Data from cell death assays and western blots are representative of 3 or more biologically independent experiments, cell death data are presented as the mean ± SD of triplicate wells, n = 10,000 cells examined in three individual wells. IL-1β release, ASC percentage, and qPCR data are presented as the mean ± SD for triplicate wells from n = 4 biologically independent experiments. Two-way analysis of variance (ANOVA) was used for comparison between groups. Source data for all experiments are provided as a Source data file. See also Supplementary Figs. 2 and 3.

phosphorylation of RIPK1 at S166 were attenuated in *Zbp1*−/− BMDMs, though to a lesser extent than in the absence of TRIF (Fig. 2f). These data suggest that ZBP1 may regulate recruitment of RIPK1 into a complex (such as TNFR1-associated complex I) in which both pro-survival and pro-death modifications of RIPK1 occur. Taken together, these results indicate that although ZBP1 is not absolutely required for CASP8-mediated cell death, the efficiency of this pathway is dependent on ZBP1.

Based on the attenuation of CASP8-mediated GSDMD cleavage in *Zbp1*−/− BMDMs, we were interested to see if ZBP1 was required for inflammasome activation and IL-1β release. We have recently shown that priming with LPS prior to 5z7 treatment upregulated pro-IL-1β and NLRP3 inflammasome components, allowing for CASP8-dependent mature IL-1β release, while simultaneously delaying the onset of cell death due to upregulation of pro-survival factors such as cFLIP (cellular FLICE-like inhibitory protein)[22]. Accordingly, we pre-primed B6 and *Zbp1*−/− BMDMs with LPS for 4 h prior to the addition of LPS/5z7, and observed upregulation and cleavage of pro-IL-1β that was dependent on ZBP1 (Fig. 2g). Furthermore, LPS/5z7 treatment induced cleavage of inflammatory CASP1, which was delayed in *Zbp1*−/−, and by LPS priming (Fig. 2g). Indeed, ZBP1 deficiency decreased IL-1β release and inflammasome activation, as measured by the percentage of ASC speck-positive cells after treatment with LPS/5z7 in LPS pre-primed macrophages (Fig. 2h, i). These data supported previous reports that ZBP1 regulates NLRP3 inflammasome activation and IL-1β release in response to influenza A viral infection[20]. Decreased upregulation of inflammasome components (CASP1 and pro-IL-1β), as well as decreased RIPK1 S321 phosphorylation (Fig. 2f) in response to LPS in *Zbp1*−/− suggests that ZBP1 may be involved in TRIF-mediated signaling in response to LPS (Fig. 2g). Indeed, ZBP1 deficiency decreased TRIF, TNF, and IFNβ mRNA levels in response to LPS and LPS/5z7, indicating that ZBP1 may regulate inflammatory signaling at multiple levels (Supplementary Fig. 3a).

To confirm that TRIF and ZBP1 are functioning within the same pathway to drive CASP8-mediated cell death, we knocked down (KD) ZBP1 in *Trif*−/− and RIPK1 kinase-inactive BMDMs (Supplementary Fig. 3b). While ZBP1 KD inhibited death in B6 BMDMs, deficiency in ZBP1 provided no further attenuation of death on the *Trif*−/− and RIPK1 kinase-inactive background, indicating that ZBP1 functions within the TRIF-mediated pathway to regulate RIPK1 binding interactions or localization (Supplementary Fig. 3c). In further confirmation of the importance of ZBP1 in TRIF-mediated pro-death signaling, both TRIF and ZBP1 deficiency decreased death in response to the TLR3/TRIF ligand Poly I:C in the context of TAK1 inhibition with 5z7 (Supplementary Fig. 3d).

**ZBP1 promotes cell death via constitutive binding to RIPK1.** To better understand the mechanism of regulation of CASP8-mediated cell death by ZBP1, we immunoprecipitated ZBP1 in resting, LPS- or LPS/5z7-treated BMDMs and probed for binding

interactions with RIPK1 and CASP8 (Fig. 3a, Supplementary Fig. 4a). Surprisingly, we observed interaction of ZBP1 and RIPK1 in untreated and LPS-treated cells, which was decreased after 2 h of LPS/5z7 treatment, accompanied by the appearance of ZBP1-bound active CASP8 (Fig. 3a, Supplementary Fig. 4a), indicative of complex II assembly. Although constitutive binding of ZBP1 and RIPK1 has not been previously reported, this may elucidate findings that ZBP1-mediated necroptosis drives perinatal lethality in RIPK1RHIM/RHIM mutant mice[23,24]. These reports demonstrate that RIPK1, through RHIM-mediated interactions serves as a brake to prevent ZBP1–RIPK3 binding and necroptosis. Constitutive binding between ZBP1 and RIPK1, and the appearance of active CASP8 bound to ZBP1 after LPS/5z7 treatment supported the hypothesis that ZBP1 may regulate CASP8-mediated cell death via RHIM interactions with RIPK1. To confirm this, we generated FLAG-tagged mutant ZBP1 constructs lacking one or both RHIM domains (Fig. 3b). Reconstitution of *Zbp1*−/− BMDMs with full-length ZBP1 (FL ZBP1), or ZBP1 lacking RHIM2 (ZBP1-R2) nearly recapitulated the extent of cell death observed in B6 BMDMs (Fig. 3b, d). However, reconstitution with ZBP1 constructs lacking RHIM1 (ZBP1-R1 and ZBP1-R1/2) failed to increase death in *Zbp1*−/− BMDMs, indicating a role for RHIM1 in LPS/5z7-induced death (Fig. 3b, d). ZBP1 is also composed of Z-DNA binding domains capable of sensing DNA and RNA viruses, and driving cell death[25]. Thus, we generated FLAG-tagged mutant constructs of ZBP1 lacking one or both Z-DNA binding domains and tested their capacity to rescue death in response to LPS/5z7 in *Zbp1*−/− BMDMs (Fig. 3c, e). All mutant constructs recapitulated reconstitution with full-length ZBP1, indicating that the Z-DNA-binding domains of ZBP1 are not required for CASP8-mediated cell death. To determine if ZBP1-R1 failed to elicit cell death due to loss of binding interactions with RIPK1 or CASP8, we precipitated exogenous ZBP1 RHIM mutant proteins with FLAG antibodies in *Zbp1*−/− BMDMs (Fig. 3f, Supplementary Fig. 4b, c). Strikingly, deficiency in RHIM1 prevented ZBP1–RIPK1 binding as well as downstream CASP8 binding and activation (Fig. 3f, Supplementary Fig. 4b, c). This data was in agreement with earlier overexpression-based studies in 293T and L929 cell lines showing a requirement for RHIM1-mediated ZBP1–RIPK1 binding in the regulation of NF-κB activation[26,27]. Altogether, these findings suggest that ZBP1 promotes cell death in BMDMs by functioning as a scaffold protein rather than a sensor of nucleic acids.

**TRIF induces complex formation independently of TNFR1.** Recently, we have shown that LPS/5z7 treatment induced the formation of a complex II-like protein complex containing FADD, RIPK1, CASP8 by decreasing cellular stores of the CASP8 homolog cFLIP. Furthermore, successful initiation of this complex was required for CASP8 activation and subsequent pyroptosis[22]. Complex II formation has been most extensively studied downstream of TNF receptor 1 (TNFR1)[28]. However, lack

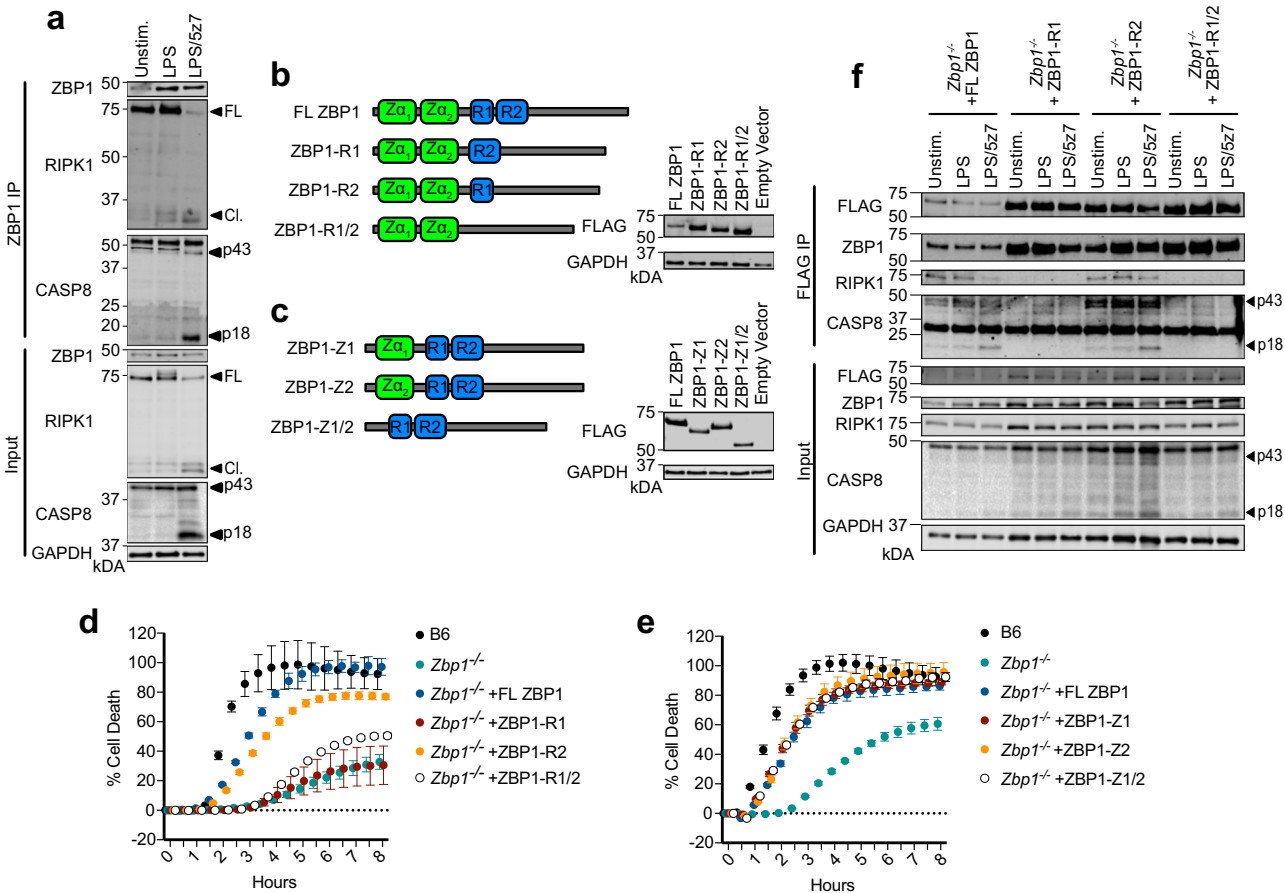

**Fig. 3 ZBP1 promotes cell death via constitutive binding to RIPK1. a** ZBP1 immunoprecipitation in B6 BMDMs stimulated as indicated for 2 h and probed for RIPK1 and CASP8. **b**, **c** Schematic of full-length and mutant ZBP1 constructs generated, and confirmation of reconstitution experiments in $Zbp1^{-/-}$ BMDMs. **d**, **e** Cell death over time as measured by propidium iodide incorporation in $Zbp1^{-/-}$ BMDMs reconstituted with indicated constructs and treated with LPS/5z7. **f** FLAG-specific immunoprecipitation in $Zbp1^{-/-}$ BMDMs reconstituted with indicated ZBP1 constructs, stimulated with LPS or LPS/5z7 for 2 h and probed for ZBP1, RIPK1, and CASP8. Data from cell death assays and western blots are representative of 3 or more biologically independent experiments, cell death data are presented as the mean ± SD of triplicate wells, $n = 10,000$ cells examined in three individual wells. Source data for all experiments are provided as a Source data file. See also Supplementary Fig. 4.

of TNFR1 had no effect on CASP8-mediated cell death induced by LPS/5z7 treatment as measured by PI incorporation and further confirmed by LDH release (Fig. 4a, b). Instead, LPS/5z7-induced death was highly dependent on TRIF, TRIF-associated adaptor (TRAM)[29], and the LPS-co-receptor CD14 (Fig. 4a) responsible for LPS-induced endocytosis of TLR4[30].

To further investigate the role of TNFR1 signaling in LPS/5z7-induced cell death, we treated B6, $Trif^{-/-}$, $Tnfr^{-/-}$, and $Zbp1^{-/-}$ BMDMs with LPS/5z7 in the presence of TNF-neutralizing antibodies (αTNF) (Fig. 4c). While αTNF had no effect in B6 and $Tnfr^{-/-}$, αTNF treatment resulted in a modest, yet statistically significant decrease in death in $Trif^{-/-}$ and $Zbp1^{-/-}$ BMDMs, indicating that TNFR1 signaling may contribute to cell death in the absence of sufficient TRIF-mediated signaling (Fig. 4c). However, the physiological significance of this modest decrease in cell death remains to be further explored. In confirmation of the efficacy of the TNF-neutralizing antibody in inhibition of TNF signaling, treatment with αTNF abrogated 5z7-induced cell death, which we have previously shown to be TNF-dependent[11] (Fig. 4d).

In the context of TLR4 signaling, CD14 serves dual functions as (1) an LPS-co-receptor important for TLR4-mediated responses to LPS and (2) a cell surface protein that regulates endocytosis of TLR4 to promote TRIF-mediated responses[30,31]. In order to decouple the requirement for these two functions in

the context of LPS/5z7-induced cell death, we treated B6 and $Cd14^{-/-}$ BMDMs with the endocytosis inhibitor Dynasore (Fig. 4e). Treatment of B6 BMDMs with Dynasore recapitulated the phenotype of $Cd14^{-/-}$ BMDMs, suggesting that endocytosis is required for LPS/5z7-induced cell death (Fig. 4e).

Lack of TRIF-specific antibodies complicated our ability to specifically observe TRIF localized within the complex. However, we were able to assess the relative contribution of TRIF and TNFR1 to complex formation using $Trif^{-/-}$ and $Tnfr^{-/-}$ BMDMs (Fig. 4f–h). TNFR1 had no impact on complex formation, as RIPK1 and CASP8 bound to FADD comparably to B6 in $Tnfr^{-/-}$ (Fig. 4h). In contrast, TRIF was absolutely required for complex formation, and CASP8 was not activated under the same conditions in $Trif^{-/-}$ BMDMs (Fig. 4g). To reflect the fact that TRIF, and not TNFR1 signaling was responsible for complex formation and activation of CASP8-mediated cell death, we termed this complex the TRIFosome[32]. In addition to the role of TRIF in TRIFosome formation and cell death, TRIF but not TNFR1 was required for LPS/5z7-induced IL-1β release (Fig. 4i). The abrogation of IL-1β release in $Trif^{-/-}$ BMDMs was even greater than in the absence of ZBP1, supporting the role for ZBP1 downstream of TRIF. LPS/5z7-induced IL-1β release was dependent on the NLRP3 inflammasome, as deficiency in NLRP3 or treatment with the NLRP3 inhibitor MCC950 largely attenuated IL-1β release (Fig. 4i).

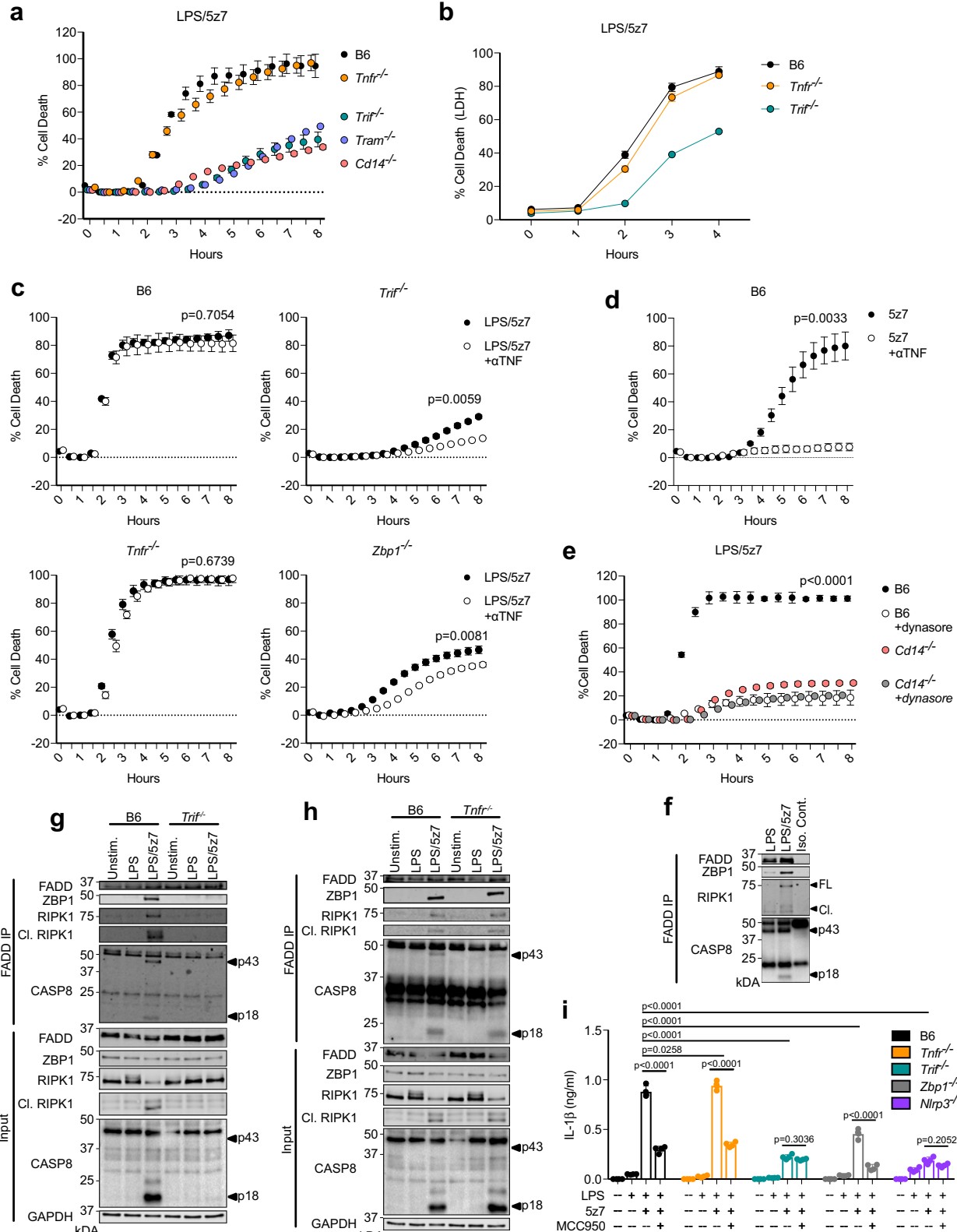

**ZBP1 and RIPK1 kinase activity regulate TRIFosome formation.** Interestingly, binding of ZBP1 to the TRIFosome indicated that ZBP1 might regulate the extent of CASP8-mediated cell death by controlling complex formation (Fig. 4g, h). To begin to address this question, we performed a time course of ZBP1-specific immunoprecipitations, and probed for critical TRIFosome components RIPK1 and CASP8 (Fig. 5a). As before, ZBP1 and RIPK1

were constitutively bound, with loss of ZBP1–RIPK1 interactions occurring at later time points after LPS/5z7 treatment (Fig. 5a), likely due to degradation of RIPK1 by CASP8 in complex, as both RIPK1 loss, and the presence of cleaved CASP8 were inhibited by treatment with zVAD and Nec-1 (Fig. 5a). This is in agreement with decreases in total RIPK1 levels that we have seen in response to LPS/5z7 treatment (Figs. 1d, 2f). In addition, we observed RIPK1

**Fig. 4 TRIF induces complex formation independently of TNFR1. a, b** Cell death over time in indicated BMDMs stimulated with LPS/5z7 as measured by **a** propidium iodide (PI) incorporation or **b** LDH release. **c–e** Cell death over time as measured by PI incorporation in indicated BMDMs treated with **c** LPS/ 5z7+/− TNF-neutralizing antibody, **d** 5z7+/− TNF-neutralizing antibody or **e** Dynasore endocytosis inhibitor. **f–h** FADD immunoprecipitation in **f** B6, **g** Trif$^{-/-}$, and **h** Tnfr$^{-/-}$ BMDMs stimulated as indicated and probed for ZBP1, RIPK1, and CASP8. **i** IL-1β release after 6 h in indicated BMDMs treated with LPS, LPS pre-primed LPS/5z7, or LPS pre-primed LPS/5z7 in the presence of NLRP3 inflammasome inhibitor MCC950. Data from cell death assays and western blots are representative of 3 or more biologically independent experiments, cell death data are presented as the mean ± SD of triplicate wells, $n = 10{,}000$ cells examined in three individual wells. IL-1β release data are presented as the mean ± SD for triplicate wells from $n = 4$ biologically independent experiments. Two-way analysis of variance (ANOVA) was used for comparison between groups. Source data for all experiments are provided as a Source data file.

phosphorylated at S321 and S166 bound to ZBP1, suggesting that these modifications occur within the complex, and supporting our hypothesis that ZBP1-mediated recruitment of RIPK1 regulates RIPK1 post-translational modifications within the TRIFosome (Fig. 5a). However, while we were able to detect ZBP1 bound in FADD immunoprecipitations (Fig. 4f, g), we were unable to detect FADD bound to ZBP1, indicating that ZBP1 interactions with the TRIFosome were highly transient. To address this, we performed FADD immunoprecipitations to assess the requirements for ZBP1 interactions with TRIFosome components. Indeed, ZBP1, as well as RIPK1 and CASP8 co-precipitated with FADD as early as 1 h after treatment of B6 BMDMs with LPS/5z7 (Fig. 5b). However, treatment with LPS/5z7 at later time points abrogated ZBP1:FADD binding, thus confirming a crucial role of ZBP1 specifically in the initiation of complex formation (Fig. 5b). Interestingly, stimulation of BMDMs with LPS alone preserved binding of ZBP1 to FADD, supporting further investigation of this complex in the context of a strictly pro-inflammatory response. Furthermore, this LPS-induced ZBP1:FADD binding was preserved and enhanced in RIPK1 kinase-inactive BMDMs (Fig. 5b). Conversely, binding of all TRI-Fosome components in response to LPS/5z7 was abrogated in RIPK1 kinase-inactive BMDMs (Fig. 5b), indicating that kinase activity of RIPK1 was required for pro-death TRIFosome formation. RIPK1 kinase activity was not important for constitutive ZBP1–RIPK1 binding, indicating that RIPK1 kinase activity is required for TRIFosome formation downstream of ZBP1 (Fig. 5c). While caspase inhibition had no effect on RIPK1 binding to ZBP1 (Fig. 5a), CASP8 deficiency decreased RIPK1 and ZBP1 binding to FADD at all time points, demonstrating that the presence of CASP8 likely stabilizes, but is not strictly required for complex formation (Fig. 5d). Finally, ZBP1 deficiency delayed TRIFosome formation and decreased the efficiency of RIPK1 binding and downstream CASP8 activation (Fig. 5e).

**The TRIFosome regulates cell death in response to Yersinia.** Recent reports have demonstrated a role for TNF signaling in driving CASP8 and RIPK1-mediated cell death in vitro, and promoting mouse survival in vivo in response to Y. pseudotuberculosis infection[33–35]. To reconcile our findings with these reports, we infected BMDMs with wild-type and YopJ-deficient (ΔYopJ) Y. pseudotuberculosis. Y. pseudotuberculosis infection of wild-type macrophages at both high (MOI 30) and low (MOI 7.5) concentrations induced cell death that was dependent on YopJ, the Yersinia effector protein capable of blocking TAK1 activation, as ΔYopJ Yersinia failed to induce death comparable to wild-type (WT) infection (Fig. 6a). However, Yersinia-induced cell death occurred more quickly at higher MOI (Fig. 6a). Similar to treatment with LPS/5z7, infection of macrophages at high MOI induced cell death that was dependent on TRIF, ZBP1, and the kinase activity of RIPK1, and independent of TNFR1 (Fig. 6b). However, unlike treatment with LPS/5z7 or high MOI Yersinia infection, which likely provide complete inhibition of TAK1 signaling, infection of macrophages at low MOI induced cell death that was additionally dependent on TNFR1, suggesting that in the absence of complete inhibition of pro-inflammatory

signaling, TNF signaling amplifies cell death in response to Yersinia infection (Fig. 6c). In further support of the importance of the TRIFosome in Yersinia infection, cleavage of CASP8 was inhibited nearly entirely in the absence of TRIF, and partially in the absence of ZBP1 and RIPK1 kinase activity in the context of high MOI Yersinia infection (Fig. 6d).

In agreement with incomplete inhibition of pro-inflammatory signaling in response to low MOI Yersinia infection, infection of macrophages at 7.5 MOI induced IL-1β release that was dependent on TRIF, TNFR1, ZBP1, and RIPK1 kinase activity (Fig. 6e). However, at 30 MOI, Yersinia infection failed to induce IL-1β release, likely due to complete inhibition of pro-IL-1β synthesis at higher MOI (Fig. 6e). In support of this, priming of macrophages with LPS prior to high MOI Yersinia infection rescued IL-1β release. Similar to cell death, IL-1β release in response to high MOI Yersinia infection was independent of TNFR1 (Fig. 6e). Taken together, these results suggest that TRIFosome components are highly necessary for cell death and IL-1β release in response to Yersinia infection, and that TNFR-signaling amplifies these processes when pro-inflammatory signaling is not completely inhibited (low MOI infection), and onset of death is delayed.

## Discussion

These results support a model in which, upon LPS induction in the context of TAK1 inhibition via 5z7 or Yersinia infection, CD14-dependent translocation of TLR4 to the endosome[30,31] permits TRIF binding to ZBP1, possibly via RHIM domain interactions. This binding recruits RIPK1, due to its engagement in constitutive RHIM:RHIM interactions with ZBP1, followed by recruitment of FADD and CASP8 downstream of TRIF (Fig. 6f). This model is in agreement with the importance of TRIF and ZBP1 for S166 phosphorylation of RIPK1, promoting the kinase function of RIPK1 required for complex formation with FADD and CASP8. Furthermore, ZBP1-dependent recruitment of RIPK1 and CASP8 to the TRIFosome is required for inflammasome activation and IL-1β release.

Despite the lack of TRIF-specific antibodies, as well as the robust cytotoxicity induced by exogenous TRIF expression[36], the striking dependence on TRIF, and relative independence from TNFR1 signaling, supported defining this ZBP1–RIPK1–FADD–CASP8-containing complex as the TRIFosome. Inter-estingly, in the context of insufficient TRIF signaling, or when pro-inflammatory signaling is only partially inhibited, as is the case with low MOI Yersinia infection, signaling through TNFR1 seems to supplement TRIF-mediated cell death and IL-1β release. In this regard, it would be interesting to assess the importance of TRIFosome formation in the context of the recently characterized CASP8-mediated cell death that occurs in response to TNF and TAK1 inhibition[13]. Although RHIM-mediated interactions between ZBP1 and RIPK1 have likely been assumed by many in the field, until now, the evidence for these interactions was largely based on overexpression studies in 293T and L929 cell lines, and had not been shown in pri-mary cells[26,27]. In addition, these RHIM-mediated interac-tions between RIPK1 and ZBP1 were mainly shown to regulate

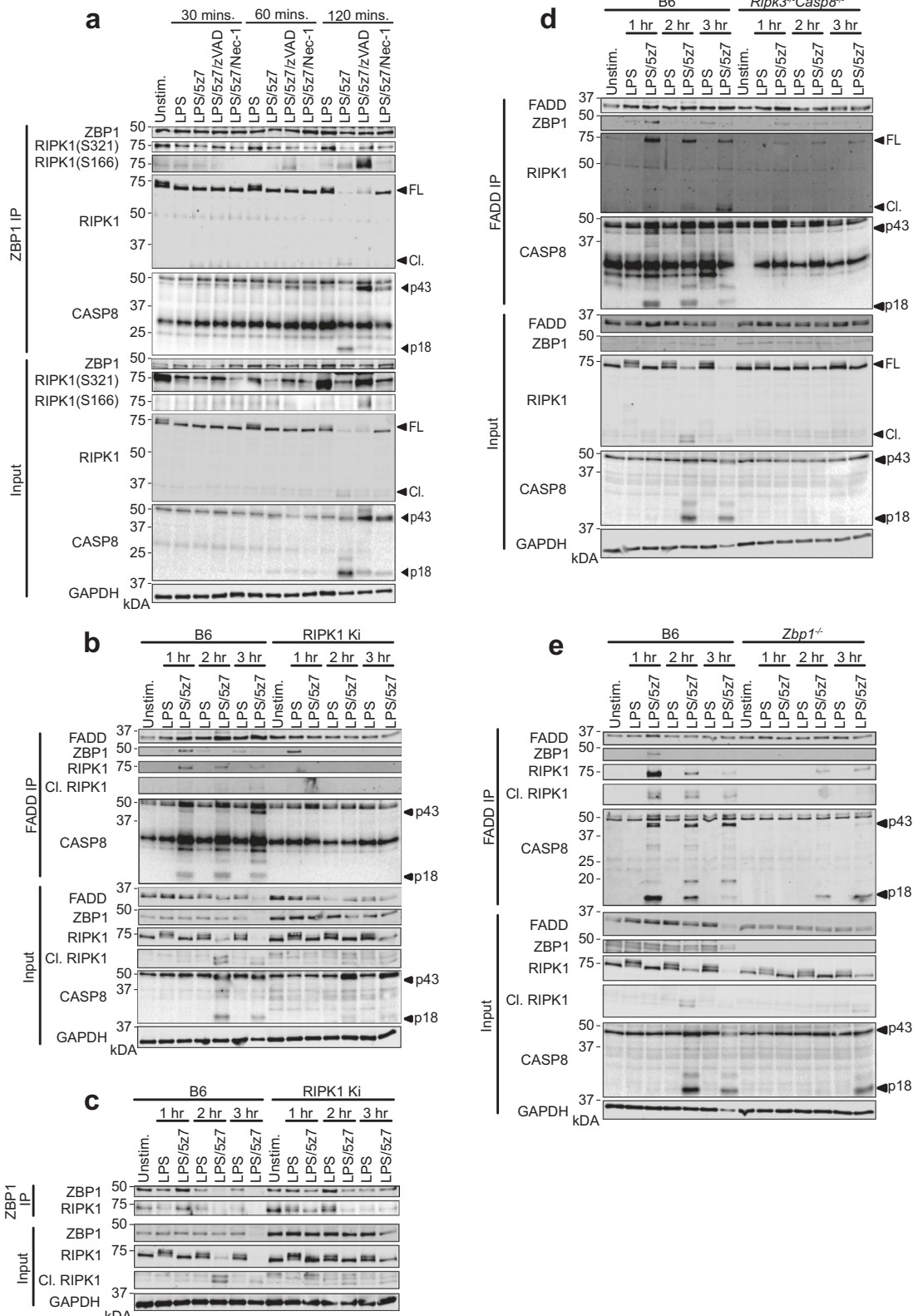

**Fig. 5 ZBP1 and RIPK1 kinase activity regulate TRIFosome formation. a** ZBP1 immunoprecipitation in B6 BMDMs stimulated as indicated and probed for RIPK1 and CASP8. **b** FADD and **c** ZBP1 immunoprecipitations in B6 and RIPK1 kinase-inactive BMDMs stimulated as indicated and probed for ZBP1, RIPK1, and CASP8. **d**, **e** FADD immunoprecipitation in B6, **d** *Ripk3*⁻/⁻*Casp8*⁻/⁻, or **e** *Zbp1*⁻/⁻ BMDMs stimulated as indicated and probed for ZBP1, RIPK1, and CASP8. Data from western blots are representative of 3 or more biologically independent experiments. Source data for all experiments are provided as a Source data file.

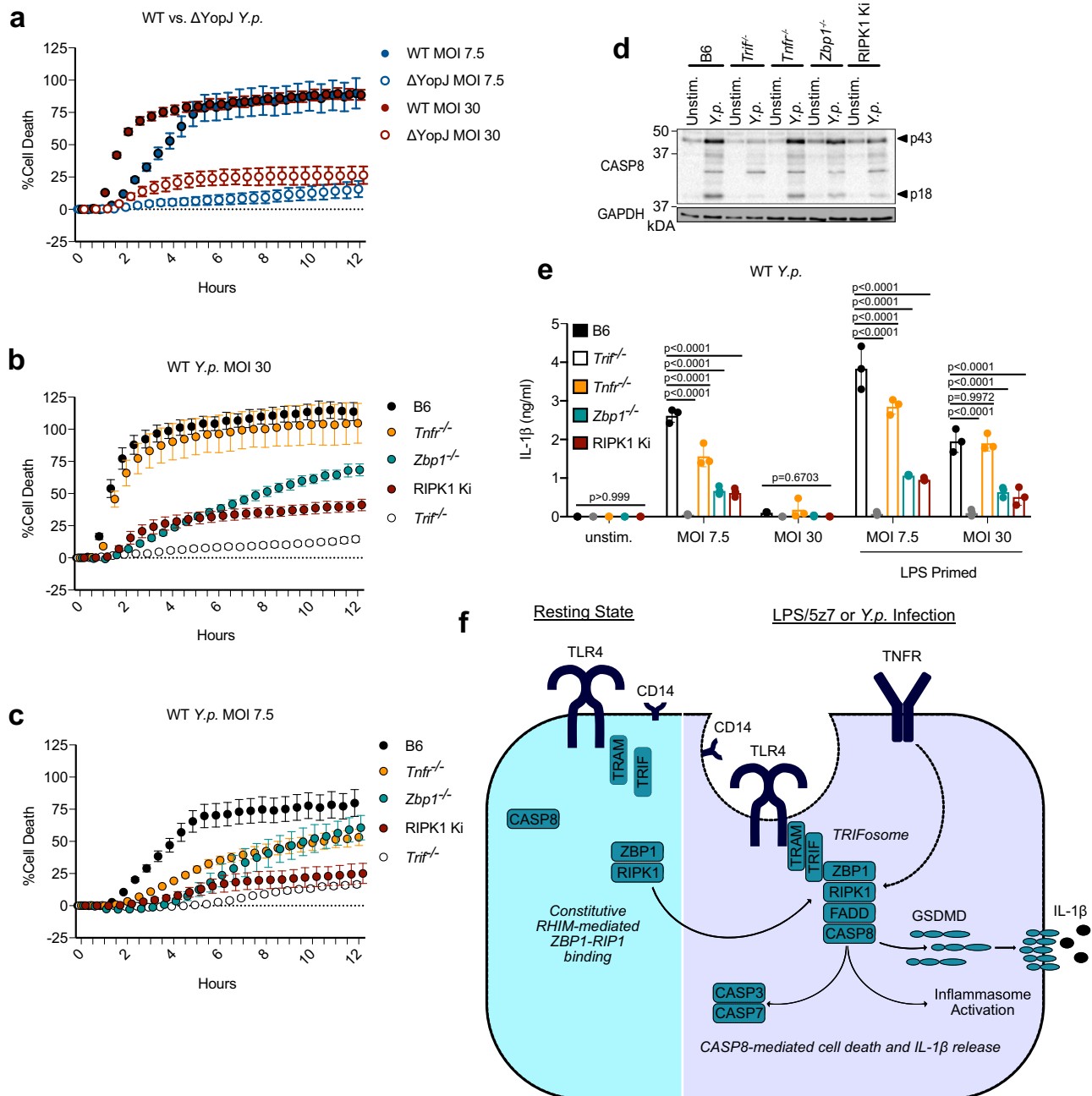

**Fig. 6 The TRIFosome regulates cell death in response to *Yersinia*. a** Cell death as measured by propidium iodide (PI) incorporation in B6 BMDMs infected with wild-type or YopJ-deficient (ΔYopJ) *Y. pseudotuberculosis* at indicated MOI. **b**, **c** Cell death as measured by PI incorporation in indicated BMDMs infected with wild-type *Y. pseudotuberculosis* at **b** 30 MOI or **c** 7.5 MOI. **d** CASP8 cleavage in indicated BMDMs infected with wild-type *Y. pseudotuberculosis* at 30 MOI for 2 h. **e** IL-1β release after 6 h of wild-type *Y. pseudotuberculosis* infection at 7.5 or 30 MOI in indicated BMDMs that were primed or unprimed with LPS for 4 h prior to infection. **f** Model of ZBP1-regulated TRIFosome formation leading to CASP8-mediated pyroptosis and IL-1β release. Data from cell death assays and western blots are representative of 3 or more biologically independent experiments, cell death data are presented as the mean ± SD of triplicate wells, *n* = 10,000 cells examined in three individual wells. IL-1β release data are presented as the mean ± SD for triplicate wells from *n* = 3 biologically independent experiments. Two-way analysis of variance (ANOVA) was used for comparison between groups. Source data for all experiments are provided as a Source data file.

pro-inflammatory NF-κB activation, while recent studies of the pro-death role of ZBP1 have focused extensively on ZBP1–RIPK3-mediated necroptosis in response to viral infection, which is dependent on ZBP1 Z-DNA binding domains[17,20,25,37]. Therefore, we have identified a novel mechanism through which ZBP1–RIPK1 RHIM-mediated interactions drive not only pro-inflammatory signaling, but also rapid CASP8-mediated cell death in response to bacterial

infection. How ZBP1 promotes CASP8-mediated cell death has yet to be fully characterized, but its contribution to complex formation appears to facilitate interactions between the RHIM domain-containing proteins RIPK1 and TRIF, thereby initiating activation of CASP8. Due to the fact that binding of ZBP1 to FADD seems to be highly transient relative to other TRIFosome interactions, with ZBP1 only detectable at very early time points after LPS/5z7 treatment, it is tempting

to postulate that ZBP1 may serve a shuttle function, initiating complex formation by providing cellular stores of RIPK1 to the TRIFosome.

The ability to engage CASP8-mediated cell death pathways has been shown to be critically important for host survival in response to *Yersinia pestis* infection in vivo, with *Ripk3*−/−*Casp8*−/− mice rapidly succumbing to infection due to uncontrolled bacterial growth and decreased inflammatory cytokine production, including IL-1β[6]. Furthermore, *Zbp1*−/− mice showed decreased rates of recovery in response to Influenza A infection due lack of immune cell infiltrates and defects in cell death and viral clearance[20]. Due to the fact that Influenza A-induced cell death and inflammasome activation are dependent on CASP8 and the RHIM domain-containing proteins RIPK1 and RIPK3[20], it is likely that RHIM-mediated interactions, similar to those explored here are responsible for the regulatory role of ZBP1 in the context of Influenza A infection. Our findings, pointing toward a TRIF–ZBP1–RIPK1–CASP8-mediated axis of regulation for cell death and inflammatory signaling are in agreement with in vivo responses in these two infection models, which in both cases are critical for host survival. Therefore, identification of ZBP1-regulated TRIFosome formation, CASP8 activation, and IL-1β production provides novel points of regulation that could be relevant in the context of various pathogenic infections.

## Methods

**Mice and macrophages**. C57BL/6 (B6), *Ticam1*−/− (*Trif*−/−, C57BL/6J-Ticam1[Lps2]/J), *Cd14*−/− (B6.129S4-*Cd14*[tmFrm]/J), *Myd88*−/− (B6.129P2(SJL)-*Myd88*[tm1.1Defr]/J), *Tnfr*−/− (C57BL/6-*Tnfrsf1a*[tm1Imx]/J), and *Nlrp3*−/− (B6.129S6-*Nlrp3*[tm1Bhk]/J) mice were obtained from The Jackson Laboratory. *Ifnb*−/− (C57BL/6 background) were a gift from Dr. S. Vogel. Mice were housed according to protocols approved by the Tufts University Medical School Animal Care and Use Committees. Mice were housed at ambient temperature and humidity in ventilated caging systems on a 12 h light cycle. All mice were housed in a specific pathogen-free facility. No more than 4 mice were housed in a single cage. Femurs from *Ticam2*−/− (*Tram*−/−, C57BL/6 background) mice were generously donated by Dr. L. Li. Femurs from *Ripk3*−/− (C57BL/6 background) and *Gsdmd*−/− (C57BL/6 background) mice were donated by Dr. K. Fitzgerald, and were generated by Dr. V. Dixit. Ripk1[K45A/K45A] (RIPK1 Ki, C57BL/6 background) mice were provided by Dr. A. Degterev. Femurs from *Ripk3*−/−*Casp8*−/− (C57BL/6 background) mice were donated by Dr. K. Fitzgerald, and were originally generated by Dr. D. Green. Femurs from *Zbp1*−/− mice (C57BL/6 background) were generously donated by Dr. S. Balachandran. Bones from *Casp3*−/−*Casp7*−/− (C57BL/6 background) were generated and provided by Dr. A. Rongvaux. Bone marrow was isolated from the long bones of 6–12-week-old male and female mice, propagated in RPMI containing 20% FBS, 2% Pen-Strep, and 30% L cell supernatant on non-tissue culture-treated Petri dishes. Once differentiated, BMDMs were plated for experiments at a density of $1 \times 10^6$ cm[2] in RPMI containing 20% FBS and 2% Pen-Strep. Immortalized *Ripk3*−/−*Casp8*−/−, *Zbp1*−/−, and B6 BMDMs were generated and donated by Dr. K. Fitzgerald.

**Reagents**. Lipopolysaccharide (LPS) *Escherichia coli* 011:B4 (10 ng/ml, L4391), 5Z-7-Oxozeaenol (5z7, 125 nM, O9890), and Necrostatin-1 (Nec-1, 10 μM, N9037) were purchased from Sigma. zVAD.fmk was purchased from Millipore (219007) and used at 50 μM. Recombinant mouse IFNβ (10 I.U., 12405-1) was purchased from PBL Assay Science. Blocking antibody to mouse IFNAR (MAR1-5A3, 20 μg/ml, 561183) and control IgG was purchased from BD Pharmingen. Dynasore endocytosis inhibitor was purchased from Sigma-Aldrich (D7693) and used at 20 μM 30 min prior to indicated stimulations. Mouse TNF-neutralizing antibody was purchased from Cell Signaling Technology (11969S) and used at a concentration of 10 ng/ml. NLRP3 inflammasome inhibitor MCC950 (inh-mcc) was purchased from InvivoGen and used at 1 μg/ml. Propidium iodide (10 μg/ml, P3566) was purchased from Invitrogen. Annexin V, Alexa Fluor 350 conjugate (A23202) was purchased from Invitrogen.

**Kinetic microscopy**. Kinetic macrophage imaging assays were performed using the Cytation3 automated microscope, and built-in environmental control maintained 37 °C, 5% $CO_2$ for the duration of the assay. Macrophages were seeded on 1.17-mm-thick glass-bottom imaging plates at a density of $1 \times 10^6$ cm[2] in RPMI. Cells were imaged at 30-min intervals at ×4 magnification to capture ~5000 cells/field of view. Propidium iodide (PI) incorporation was detected at 617 nm, and PI+ nuclei were counted. Wells treated with 0.1% Triton X-100 were used as controls for 100% cell death. For Annexin V/PI stained ×20 images as in 2G, macrophages were seeded in RPMI buffered at pH 7.2–7.5 (1 mM HEPES) and supplemented with 2 mM $CaCl_2$ to allow for Annexin V

binding. Cells were imaged at 30-min intervals at ×20 magnification, PI incorporation was detected at 617 nm, and Annexin V was detected at 350 nm.

**LDH cytotoxicity assay**. At indicated time points after stimulation with LPS/5z7, cell supernatants were collected and LDH release was measured using CyQuant LDH Cytotoxicity Assay kits (C20300) according to the manufacturer's instructions. Absorbance at 490 and 689 nm was measured using The Cytation3 automated microscope.

**Immunoblotting**. After indicated treatments, cells were lysed in 1X Laemmli Buffer containing 5% β-mercaptoethanol, boiled for 15 min and incubated on ice for 15 min. For indicated conditions, proteins were precipitated from cell supernatants by methanol–chloroform extraction, and precipitated proteins were processed as above prior to loading and running on SDS-PAGE gels. Primary antibodies against Casp8 (8592), Casp3 (9665), Casp7 (9492), Casp9 (9508), Casp1 (24232), Casp11 (14340), RIPK1 (3493), pRIPK1 (Ser166) (31122), pRIPK1 (Ser321) (38662), IL-1β (12242), FLAG (8146), and GAPDH (2118) were purchased from Cell Signaling Technologies. ZBP1 antibody (AG-20B-0010-C100) was purchased from Adipogen. GSDMD antibody (ab209845) was purchased from Abcam. FADD antibody (05-486) was purchased from Millipore Sigma. All primary antibodies were used at a dilution of 1:1000. Secondary antibodies, anti-rabbit IgG (H + L) (DyLight™ 800 4X PEG Conjugate) (5151), anti-mouse IgG (H + L) (DyLight™ 800 4X PEG Conjugate) (5257), and HRP-linked anti-rat IgG antibody (7077) were purchased from Cell Signaling Technologies. Fluorescent secondary antibodies were used at a dilution of 1:30,000 and HRP-based secondary antibodies were used at a dilution of 1:2000. Uncropped western blots are available in the Source data file.

**Next-generation RNA sequencing**. Total RNA was isolated from unstimulated and LPS/5z7 stimulated (1 h) B6, *Trif*−/− and *Myd88*−/− BMDMs using TRIzol. A TrueSeq kit was used to make a directional cDNA library. Seventy-five bp reads from cDNA libraries were generated on MiSeq (Illumina) and aligned using TopHat2 and Cufflinks software. Log-transformed values of genes that were upregulated after LPS/5z7 stimulation over unstimulated (>1.7-fold) were compared among B6, *Trif*−/− and *Myd88*−/−, and displayed using a Venn Diagram. Genes upregulated in B6 and *Myd88*−/− but not *Trif*−/− were clustered by GeneOntology (GO).

**RNA preparation and analysis**. Total RNA was isolated from $5 \times 10^5$ cells after indicated treatments using TRIzol (Invitrogen), following the manufacturer's instructions. cDNA was synthesized by reverse transcription, performed using M-MuLV reverse transcriptase, RNase inhibitor, random primers, and dNTP mix (New England BioLabs). cDNA was analyzed for relative mRNA levels using SYBR Green (Applied Biosystems), and gene-specific primers (Table S1). ActB was used to normalize mRNA levels, and post amplification melting curve analysis was performed to confirm primer specificity.

**ELISA**. Six hours after indicated treatments, cell supernatants were collected and cytokine secretion was measured by ELISA. Murine IL-1β DuoSet ELISA Kits (DY401) were used according to the manufacturer's instructions.

**ASC speck quantification and high magnification imaging**. To quantify ASC+ cells, macrophages were seeded at a density of $1 \times 10^6$ cm[2] in RPMI on 1.17-mm-thick glass-bottom imaging plates. As indicated, macrophages were unstimulated or pre-primed with LPS for 4 h prior to treatment with LPS or LPS/5z7 for 2 h. Cells were fixed in 4% paraformaldehyde for 15 min, blocked in 1× PBS (5% FBS, 0.3% Triton X-100), and incubated overnight with anti-ASC antibody (Cell Signaling Technologies, 67824), followed by a 2-h incubation in the presence of Alexa Fluor 488-conjugated-Goat-anti-Rabbit IgG (Invitrogen A-11034), and Hoechst 33342 fluorescent stain. To quantify the percentage of ASC+ cells, the Cytation3 automated microscope was used to image cells at ×4 magnification to capture ~5000 cells/field of view in quadruplicate. Hoechst stain was detected at 350 nm and Hoechst+ nuclei were counted to provide total cell count. The ASC signal was detected at 488 nm and ASC specks (1–3 μm) were counted.

**Lentivirus constructs and transduction**. For ZBP1 reconstitution experiments, full-length and mutagenized versions of ZBP1 sequences were ligated into a pLEX lentiviral vector. Lentiviral particles were generated by transfection with packaging vector psPAX2 (plasmid 12260; Addgene) and the VSV-G pseudotyping vector pMD2.G (plasmid 12259; Addgene) into the 239T cell line. Generated lentiviral particles were used to transduce *Zbp1*−/− BMDMs on day 4 of differentiation, followed by puromycin selection (3 μg/ml, 48 h) starting on day 6. Mission shRNA plasmids containing hairpin sequences specific to mouse ZBP1 were identified and purchased from Sigma-Aldrich (5′-CCGGGTCCAGACAGTCCACATCAAAC TCGAGTTTGATGTGGACTGTCTGGAACTTTTTG-3′). Scramble control vector was purchased from Addgene (plasmid 1864). Lentiviral particles from shRNA plasmids were generated as described above, and used to transduce indicated BMDMs.

**ZBP1 and FADD Immunoprecipitations.** Indicated BMDMs were plated on 6-well tissue culture-treated plates, stimulated as indicated, and harvested in immuno-precipitation lysis buffer (0.5% Triton X, 50 mM Tris Base (pH 7.4), 150 mM NaCl, 2 mM EDTA, 2 mM EGTA, 1X protease inhibitor cocktail). Lysed cells were rotated for 60 min at 4 °C with intermittent vortexing, centrifuged at $5000 \times g$ for 5 min, and the supernatant was incubated with α-FADD, α-ZBP1, or α-FLAG antibody-conjugated Protein G (FADD and ZBP1-specific immunoprecipitations) or Protein A (FLAG-specific immunoprecipitations) agarose beads (Cell Signaling Technology 37478, or 9863). Samples were washed three times in immunopreci-pitation lysis buffer, and protein complexes were eluted with 1X Laemmli buffer containing 5% β-mercaptoethanol at 90 °C for 15 min.

**Yersinia growth conditions and infection.** Wild-type and ΔYopJ IP2666 *Y. pseudotuberculosis* bacterial strains were generously provided by Dr. R. Isberg. Bacteria were grown from frozen glycerol stocks on LB plates containing Irgasan (Sigma). Cultures were grown overnight at 26 °C in 2XYT broth, diluted to an $OD_{600}$ of 0.2, and grown at 26 °C for 2 additional hours prior to a shift to 37 °C for 2 h. Macrophages were infected at MOI 7.5 or 30 CFU/cell as indicated.

**Quantification and statistical analysis.** Error bars in ELISA experiments represent the standard deviation of three independent experiments. Data from kinetic cyto-toxicity and ASC⁺ cell quantification experiments are representative of three or more experiments, and error bars represent the standard deviation between triplicate sam-ples. Immunoblots are representative of three or more independent experiments. Significance was determined using a one-way or two-way ANOVA as appropriate: ns (non-significant) $p > 0.05$; *$p < 0.05$; **$p < 0.01$; ***$p < 0.001$; ****$p < 0.0001$).

**Reporting summary.** Further information on research design is available in the Nature Research Reporting Summary linked to this article.

## Data availability

Next-generation RNA sequencing data that support the findings of this study have been deposited in the GenBank GEO Database with primary accession code GSE83885. All additional data are available within the paper and its Supplementary Information and Source data files. Reported interferon-stimulated genes within our dataset were identified using the INTERFEROME database (http://www.interferome.org/interferome/site/showCitation.jspx). Source data are provided with this paper.

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

## Acknowledgements

We thank Dr. K. Fitzgerald and Dr. S. Balachandran for sharing various mouse strains for this study. In addition, we thank Dr. S. Balachandran for reading this manuscript and offering helpful comments and expertise. We thank Dr. R. Isberg for sharing the *Y. pseudotuberculosis* strains used in this study. We thank A. Tai and the Tufts University Genomics Core for help with RNA sequencing and data analysis. The work was sup-ported by the NIH grants AI056234 to A.P., R21NS111395 and R01AI144400 to A.D.

## Author contributions

Conceptualization, A.P. and H.I.M.; validation, H.I.M. and W.M.C.; formal analysis, H.I.M. and W.M.C.; investigation, H.I.M., W.M.C., Z.M., I.S., A.G., and V.I; writing, A.P., H.I.M., W.M.C., and A.D.; visualization, H.I.M. and W.M.C.; supervision, A.P., A.D.; project administration, A.P.; funding acquisition, A.P.

## Competing interests

The authors declare no competing interests.
