## [Peer Review File · Nature Communications]

REVIEWER COMMENTS

Reviewer #1 (Remarks to the Author):

The manuscript by Muendlein and colleagues describes a novel ZBP1-containing complex to further characterize the mechanisms of Casp8-mediated pyroptosis. A number of recent studies have found that the canonical cell death types (apoptosis, pyroptosis, necroptosis) and the proteins that carry them out are interlinked and not as separate as they were previously characterized. This group and others have previously demonstrated a Casp8-mediated pyroptosis in the context of *Yersinia* infection, which involves Casp8-dependent Gasdermin D cleavage and IL-1 β release. However, the specific molecular mechanism of this cell death pathway has not been elucidated. Here, using primary bone marrow macrophages with a combination of cell death assays, immunoblots, and immunoprecipitations, Muendlein et al show that this cell death is mediated by TNFR-independent, TRIF-dependent complex that includes FADD, Casp8, and RIP1, which they term the 'TRIFosome.' Additionally, they find that ZBP1 is constitutively bound to this complex and is required for efficient induction of Casp8-mediated pyroptosis. This is a novel finding that sheds further light on the Casp8-mediated pyroptosis as well as on a previously undescribed role for ZBP1 during cell death. The finding that this pathway is independent of TNF signaling is intriguing but also a bit surprising, given that caspase-8-dependent cell death in response to *Yersinia* involves both TRIF and TNFR acting in concert. Notably, this manuscript does not assess Casp8-induced pyroptosis under physiological conditions and solely base their findings on systems involving pharmacologically-induced cell death (TAK1 inhibitor+LPS). This inhibitor treatment mimics the function of pathogen virulence factors that interfere with immune signaling, and therefore it would be important to know whether the role of ZBP1 here is specific to the pharmacological treatment or also has a physiological parallel. A number of critical controls appear to be absent in several panels and a few points would be important for this manuscript to clarify. I fully appreciate that during these unusual times the authors have a claim to the argument that addressing these questions is not possible or beyond the scope of this work, but I nevertheless believe these are important points to address in order to validate the main thesis of this paper.

1. Western blots and immunoprecipitations form a major component of this paper. The authors use western blots to demonstrate activation of RIP1, a number of caspases, and Gasdermin D. They use immunoprecipitations to demonstrate their central point that recruitment and activation of 'TRIFosome' components, ZBP1, RIP1, FADD, and Casp8, mediate efficient pyroptosis during pathogenic blockade. The authors utilize a number of immunoprecipitation-western blots to demonstrate their central point that ZBP1, RIP1, FADD, and Casp8 form a complex that mediates efficient pyroptosis during pathogenic blockade. However, these figures (Fig 3 and 4) are missing input blots, loading controls, and isotype pulldown controls. Without these key basic controls, it is not possible to interpret the data and the central conclusion that ZBP1 and FADD "binding is enhanced" (Page 13 Lines 19-20) cannot be concluded. In addition, a few more controls are missing. In Fig 4D, an Unstim. control is missing. It would be important to know whether Casp8 does or does not bind to ZBP1 in the absence of stimuli (presumably not? But regardless important to know). Additionally, Fig 2D, 3A, 4D all show cleaved Casp8, but they should include full-length Casp8 in order to visualize the kinetics of Casp8 processing. Blots with total RIP1 should also display cleaved RIP1. In a similar vein, in Fig 4D, there is a double band visible for full-length RIP1. Can the authors please explain/clarify this? Without understanding this, it is difficult to interpret the authors' claim that there is a loss of ZBP1-RIP1 interactions over time (Page 13 Line 5).

2. In order to induce cell death, the authors use only a pharmacological approach, which comes with its own set of caveats. It would be important for the authors to validate their findings using *Yersinia* infection in different knockout BMDMs (particularly *Zbp1*) in vitro and perform propidium iodide uptakes and westerns to verify their findings from LPS/5z7.

3. Does ZBP1^{-/-} influence basal expression of TRIF and/or RIP1? It would be important to

demonstrate that the expression levels are constant using qPCR and/or western blot in the supplemental, just to ensure that this does not explain your phenotypes.

4. The authors reference studies that have demonstrated the role of ZBP1 in mediating necroptosis (Page 3). However, it should be mentioned that it has already been shown that RIP1 acts as a brake to prevent ZBP1 from binding RIPK3 to induce necroptosis (Newton 2016, Lin 2016, Ingram 2019). This is why in the absence of RIP1, ZBP1 drives necroptosis and organismal lethality during development. It's not really a surprise that RIP1 and ZBP1 are constitutively bound. What is truly novel here is that ZBP1 is required for efficient pyroptosis in the context of bacterial signaling blockade. The authors should discuss this aspect of the literature in a more open way as the constitutive binding of RIP1 and ZBP1 has been reported.

5. In Fig S1B, a control for the IFNAR antibody is missing. It would also be advisable to perform this experiment in IFNAR^{-/-} cells and test this in other IFNs.

6. The authors utilize propidium iodide uptake to assay for cell death. Although this is a well-established measure of cell death, it is still generally validated by other measures of cell death such as LDH release, in order to establish that cell death is actually occurring. It would be advisable to do so here. Additionally, it would be best to include in the figure legends that cell death was measured by propidium iodide uptake.

7. The authors find that this ZBP1-dependent pathway of casp8-mediated pyroptosis is independent of TNFR. This makes sense given that TAK1 blockade would be expected to limit expression of TNF. Nevertheless, previous findings have found that for Yersinia infection (which is a physiological inhibitor of TAK1 that induces casp8 activation), synergy between TNFR1 and TRIF was necessary to induce maximal cell death. How do the authors account for this apparent difference? In this manuscript, cell death is measured as PI uptake. However, given the findings that pore formation can take place without cell death ('hyperactivation') it is important to know whether death in the TNFR1 knockout cells is indeed equivalent, or whether a shift to hyperactivation occurs in the absence of TNFR1 that ostensibly maintains PI uptake and caspase cleavage.

Typos/minor errors:

- Fig 1B: text says Casp1, it seems that this should be Casp8 (?)
- Page 7 Line 16: It says Rip3Casp8^{-/-} but it should say Rip3^{-/-}Casp8^{-/-}
- Fig 2D: Casp7 WCL: which band is full length? There are 2 bands shown, and the arrow falls between them.
- Page 14 Line 3 typo: It says RIP instead of RIP1

Reviewer #2 (Remarks to the Author):

This manuscript reports that ZBP1 is important for LPS-TRIF killing responses upon LPS treatment and TAK1 inhibition, and that ZBP1 is present within TRIF complexes, possibly via TRIF RHIM-mediated recruitment, which thereby triggers RIPK1-caspase-8. Although these findings are of interest to the field the data presented often does not support the conclusions drawn, controls are often lacking (see specific comments below), and whether the results of TAK1 inhibition mirrors Yersinia infection, as the authors allude to, is not tested. Other comments:

1. Westerns lack molecular weight markers making their accuracy difficult to assess.
2. If ZBP is required for RIPK1 recruitment to TRIF complexes, as the authors conclude, then is RIPK1 S166 phosphorylation and degradation upon LPS/TAK1 inhibitor treatment blocked to the

same extent as shown for TRIF deficient cells (Figure 1)?

3. Figure 1A. The authors note "we identified the TLR4 adaptor TRIF as an important mediator of pyroptosis (Figure 1A).", as their manuscript title also alludes to. It would benefit this take-home conclusion if the authors directly compare, at the same time, LPS/TAK-1 inhibitor killing in TRIF vs GSDMD KO BMDMs, as TRIF can also engage apoptotic caspases, as the authors go on to demonstrate in Figure 1B, and I would be surprised if the death is not a combination of both apoptosis and pyroptosis; which should be clearly resolved for readers and the text adjusted if this is found to be true (as hinted at by the authors own data presented in figure 2C). Moreover, relative to apoptotic caspase cleavage, the cleavage of GSDMD to the 30 kDa pore-forming fragment is not apparent in Figure 1B (that the authors indicate is representative of 3 independent experiments), which would suggest that GSDMD is not even activated.

4. Figure 1A/B. The authors data demonstrate that over-time TRIF-independent killing can take place. Does this reflect autocrine TNF-TNFR1 activation or is the TAK1 inhibitor toxic (inhibitor alone and LPS alone treatment control data is not shown)? It is also unclear how caspase-9 comes to be activated with similar kinetics to caspase-8 in LPS/TAK1 inhibitor treated cells (Figure 1B), as caspase-9 processing is a readout of mitochondrial apoptosis not typically associated with LPS killing.

5. Figure 2B/C. As noted above, it is important to define how much TRIF/ZBP1 independent death is TNF driven (e.g. via neutralising TNF antibodies and/or deletion on top of TRIF KO and ZBP1 KO cells [using appropriate controls]).

6. The authors write "To further characterize cell death in Zbp1^{-/-} BMDMs, we observed that although Annexin V and propidium iodide (PI) staining was delayed in the absence of ZBP1, PI incorporation and weak Annexin V staining occurred nearly simultaneously in both B6 and Zbp1^{-/-} BMDMs, consistent with the pyroptotic mechanism of CASP8-mediated cell death (Figure S1F)." It is unclear to me how PI staining can be delayed in the absence of ZBP1 yet occur simultaneously? Do they mean that ZBP1 deficient cells eventually become PI positive? How is this "consistent with the pyroptotic mechanism of casp8-mediated cell death"?

7. Figure 2D. Caspase-8 and caspase-3 cleavage is not altered by Zbp1 loss, while pyroptotic GSDMD p30 cleavage is not significantly induced by LPS/TAK-inhibitor treatment (as per Figure 1A). Therefore this data does not agree with the authors conclusions that these cleavage events "were delayed by ZBP1 deficiency (Figures 1B, 2D)." and a "pyroptotic cell death mechanism".

8. Figure 2E-2G. Decreased IL-1 release upon casp8 deficiency may simply reflect impaired inflammasome priming (e.g. LPS-induced NLRP3 and precursor IL-1 levels), as many studies have reported. Similarly, measuring precursor and bioactive (i.e. cleaved) caspase-1 and IL-1 levels in ZBP1 KOs is required, and showing this is NLRP3 dependent (e.g. genetically or via MCC950 inhibition) is important for the authors conclusions. It is also wrong to conclude that "This decrease in inflammasome-mediated IL-1 β release was likely due to the delayed kinetics of CASP8 activation observed in Zbp1^{-/-} BMDMs." when, in fact, Figure 2D suggests equivalent, or even more, caspase-8 is cleaved in ZBP1 deficient cells.

9. Figure 3A/F and elsewhere (e.g. figure 4). Essential input and ZBP KO controls are lacking.

10. Figure 4A. As indicated above, the significant cell death observed in TRIF KO cells at later time points might still reflect TNFR1 signalling and this should be tested.

11. If, as the authors allude ZBP1 is critical for the "TRIFosome" response, do ZBP-1 deficient macrophages, like TRIF deficient cells, show reduced LPS-induced IRF/IFN β responses and inflammatory cytokine production (e.g. see DOI: 10.1126/science.1087262). Is TLR3-TRIF signalling, including TLR3-induced cytokine and killing responses also altered in ZBP-1 deficient cells? Are ZBP deficient animals resistant to endotoxic shock, akin to TRIF KO animals?

Below is our point-by-point response to the concerns of the reviewers. The reviewers' comments are highlighted in gray followed by our response.

Reviewer #1 (Remarks to the Author):

The manuscript by Muendlein and colleagues describes a novel ZBP1-containing complex to further characterize the mechanisms of Casp8-mediated pyroptosis. A number of recent studies have found that the canonical cell death types (apoptosis, pyroptosis, necroptosis) and the proteins that carry them out are interlinked and not as separate as they were previously characterized. This group and others have previously demonstrated a Casp8-mediated pyroptosis in the context of Yersinia infection, which involves Casp8-dependent Gasdermin D cleavage and IL-1 β release. However, the specific molecular mechanism of this cell death pathway has not been elucidated. Here, using primary bone marrow macrophages with a combination of cell death assays, immunoblots, and immunoprecipitations, Muendlein et al show that this cell death is mediated by TNFR-independent, TRIF-dependent complex that includes FADD, Casp8, and RIP1, which they term the 'TRIFosome.' Additionally, they find that ZBP1 is constitutively bound to this complex and is required for efficient induction of Casp8-mediated pyroptosis. This is a novel finding that sheds further light on the Casp8-mediated pyroptosis as well as on a previously undescribed role for ZBP1 during cell death. The finding that this pathway is independent of TNF signaling is intriguing but also a bit surprising, given that caspase-8-dependent cell death in response to Yersinia involves both TRIF and TNFR acting in concert. Notably, this manuscript does not assess Casp8-induced pyroptosis under physiological conditions and solely base their findings on systems involving pharmacologically-induced cell death (TAK1 inhibitor+LPS). This inhibitor treatment mimics the function of pathogen virulence factors that interfere with immune signaling, and therefore it would be important to know whether the role of ZBP1 here is specific to the pharmacological treatment or also has a physiological parallel. A number of critical controls appear to be absent in several panels and a few points would be important for this manuscript to clarify. I fully appreciate that during these unusual times the authors have a claim to the argument that addressing these questions is not possible or beyond the scope of this work, but I nevertheless believe these are important points to address in order to validate the main thesis of this paper.

We are excited to hear that the reviewer thinks our findings are novel and intriguing. As described more extensively below, in the revised version of the manuscript we believe we have addressed the reviewer concerns regarding the role of TNF signaling in cell death, the physiological relevance of our findings and the lack of specific controls.

1. Western blots and immunoprecipitations form a major component of this paper. The authors use western blots to demonstrate activation of RIP1, a number of caspases, and Gasdermin D. They use immunoprecipitations to demonstrate their central point that recruitment and activation of 'TRIFosome' components, ZBP1, RIP1, FADD, and Casp8, mediate efficient pyroptosis during pathogenic blockade. The authors utilize a number of immunoprecipitation-western blots to demonstrate their central point that ZBP1, RIP1, FADD, and Casp8 form a complex that mediates efficient pyroptosis during pathogenic blockade. However, these figures (Fig 3 and 4) are missing input blots, loading controls, and isotype pulldown controls. Without these key basic controls, it is not possible to interpret the data and the central conclusion that ZBP1 and FADD "binding is enhanced" (Page 13 Lines 19-20) cannot be concluded. In addition, a few more controls are missing. In Fig 4D, an Unstim. control is missing. It would be important to know whether Casp8 does or does not bind to ZBP1 in the absence of stimuli (presumably not? But regardless important to know). Additionally, Fig 2D, 3A, 4D all show

cleaved Casp8, but they should include full-length Casp8 in order to visualize the kinetics of Casp8 processing. Blots with total RIP1 should also display cleaved RIP1. In a similar vein, in Fig 4D, there is a double band visible for full-length RIP1. Can the authors please explain/clarify this? Without understanding this, it is difficult to interpret the authors' claim that there is a loss of ZBP1-RIP1 interactions over time (Page 13 Line 5).

We agree that input blots, loading controls and isotype pulldown controls are important for interpretation of the data presented. Accordingly, in the revised version of the manuscript, we have included input blots and loading controls for all immunoprecipitation based experiments. Additionally, we have included panels that include isotype pulldown controls in wild type (B6) macrophages for FADD, ZBP1 and FLAG-specific immunoprecipitations. For Figure 4D (now 5A) we have replaced the previous ZBP1 immunoprecipitation with another similar immunoprecipitation that includes an unstimulated control. We appreciate that blots for full-length CASP8 in our immunoprecipitations would be informative, however overlap of the antibody heavy chain band with the full-length CASP8 band often makes visualization difficult under these conditions. We hope that blots showing processing to p43 and p18 are sufficient to allow for the visualization of the kinetics of CASP8 cleavage. All blots in which cleaved RIP1 is detectable now display cleaved RIP1 in the revised manuscript. In Figure 4D (now 5A) the double band visible for full-length RIP1 is due to a phosphorylation specific band shift, we have included blots for phospho-RIP1 S166 and S321 in this panel to help demonstrate this point. The double band in total RIP1 can be attributed to phosphorylation of RIP1 at S321, which is promoted by LPS and inhibited by TAK1 inhibition with 5z7.

2. In order to induce cell death, the authors use only a pharmacological approach, which comes with its own set of caveats. It would be important for the authors to validate their findings using *Yersinia* infection in different knockout BMDMs (particularly *Zbp1*) in vitro and perform propidium iodide uptakes and westerns to verify their findings from LPS/5z7.

We agree that validating our findings in the context of *Yersinia* infection would be very informative. To that end, we have performed experiments with *Yersinia pseudotuberculosis* infection in *Trif*^{-/-}, *Tnfr*^{-/-}, *Zbp1*^{-/-} and RIP1 kinase inactive macrophages to look at cell death, CASP8 cleavage and IL-1β release. These experiments support our results with LPS/5z7 treatment, and we feel they greatly add to the physiological relevance of our report.

3. Does ZBP1^{-/-} influence basal expression of TRIF and/or RIP1? It would be important to demonstrate that the expression levels are constant using qPCR and/or western blot in the supplemental, just to ensure that this does not explain your phenotypes.

In the revised version of the manuscript, we have included western blots for RIP1 in *Zbp1*^{-/-} to show that ZBP1 does not appear to affect RIP1 levels. However, ZBP1 does appear to play some role in LPS-signaling (a topic that we are currently following up on in the lab for another report) and therefore does have some effect on RIP1 phosphorylation and TRIF mRNA levels.

4. The authors reference studies that have demonstrated the role of ZBP1 in mediating necroptosis (Page 3). However, it should be mentioned that it has already been shown that RIP1 acts as a brake to prevent ZBP1 from binding RIPK3 to induce necroptosis (Newton 2016, Lin 2016, Ingram 2019). This is why in the absence of RIP1, ZBP1 drives necroptosis and organismal lethality during development. It's not really a surprise that RIP1 and ZBP1 are constitutively bound. What is truly novel here is that ZBP1 is required for efficient pyroptosis in the context of bacterial signaling blockade. The authors should discuss this aspect of the literature in a more open way as the constitutive binding of RIP1 and ZBP1 has been reported.

We appreciate that the reviewer has identified the novelty of our findings, and in the revised manuscript we have improved the discussion of this aspect of the literature.

5. In Fig S1B, a control for the IFNAR antibody is missing. It would also be advisable to perform this experiment in IFNAR^{-/-} cells and test this in other IFNs.

In this figure, B6 cells treated with IFNAR blocking antibody were compared to B6 cells treated with IgG control, we have made this more clear in the figure. We have also included data on cell death in *Ifnar*^{-/-} macrophages that were not included in the previous version.

6. The authors utilize propidium iodide uptake to assay for cell death. Although this is a well-established measure of cell death, it is still generally validated by other measures of cell death such as LDH release, in order to establish that cell death is actually occurring. It would be advisable to do so here. Additionally, it would be best to include in the figure legends that cell death was measured by propidium iodide uptake.

In the revised version of the manuscript we have included analyses of LDH release in response to LPS/5z7 treatment in B6, *Trif*^{-/-}, *Tnfr*^{-/-} and *Zbp1*^{-/-} macrophages. We have also included the method used to measure cell death (propidium iodide incorporation vs. LDH release) in the figure legends wherever applicable.

7. The authors find that this ZBP1-dependent pathway of casp8-mediated pyroptosis is independent of TNFR. This makes sense given that TAK1 blockade would be expected to limit expression of TNF. Nevertheless, previous findings have found that for *Yersinia* infection (which is a physiological inhibitor of TAK1 that induces casp8 activation), synergy between TNFR1 and TRIF was necessary to induce maximal cell death. How do the authors account for this apparent difference? In this manuscript, cell death is measured as PI uptake. However, given the findings that pore formation can take place without cell death ("hyperactivation") it is important to know whether death in the TNFR1 knockout cells is indeed equivalent, or whether a shift to hyperactivation occurs in the absence of TNFR1 that ostensibly maintains PI uptake and caspase cleavage.

We agree that this is a very interesting question. In the revised version of the manuscript we have explored this question via our experiments with *Y. pseudotuberculosis*. We found that in the context of higher MOI *Yersinia* infections (30 MOI) cell death occurs independently of TNFR1, but is dependent on TRIF as was true for our LPS/5z7 based experiments. However, when macrophages were infected at lower MOI (7.5), cell death was slower and partially dependent on both TRIF and TNFR1, suggesting that when TNF-signaling is not completely blocked, signaling through TNFR1 amplifies TRIF-dependent death. We think these findings contribute to our nuanced understanding of the crosstalk between TRIF and TNFR1-mediated signaling as they impact cell death. Furthermore, we have performed LDH release assays in *Trif*^{-/-} and *Tnfr*^{-/-} to confirm the results from our propidium iodide incorporation assays, which show that LPS/5z7-induced cell death is not dependent on TNFR1.

Typos/minor errors:

- Fig 1B: text says Casp1, it seems that this should be Casp8 (?)
- Page 7 Line 16: It says Rip3Casp8^{-/-} but it should say Rip3^{-/-}Casp8^{-/-}
- Fig 2D: Casp7 WCL: which band is full length? There are 2 bands shown, and the arrow falls between them.
- Page 14 Line 3 typo: It says RIP instead of RIP1

These errors have been corrected in the revised manuscript.

Reviewer #2 (Remarks to the Author):

This manuscript reports that ZBP1 is important for LPS-TRIF killing responses upon LPS treatment and TAK1 inhibition, and that ZBP1 is present within TRIF complexes, possibly via TRIF RHIM-mediated recruitment, which thereby triggers RIPK1-caspase-8. Although these findings are of interest to the field the data presented often does not support the conclusions drawn, controls are often lacking (see specific comments below), and whether the results of TAK1 inhibition mirrors *Yersinia* infection, as the authors allude to, is not tested. Other comments:

We are excited to hear that our findings are of interest to the field. In the revised version of the manuscript we have worked to ensure that our conclusions are supported by the data presented and that the necessary controls are included. Additionally, we have performed experiments with *Yersinia pseudotuberculosis* infection in *Trif*^{-/-}, *Tnfr*^{-/-}, *Zbp1*^{-/-} and RIP1 kinase inactive macrophages to look at cell death, CASP8 cleavage and IL-1 β release.

1. Westerns lack molecular weight markers making their accuracy difficult to assess.

Due to the nature of our western blot imaging, molecular weight markers and protein bands were imaged on different fluorescent channels that we can visualize simultaneously for the purpose of analysis, but that we separate for publication purposes to decrease background noise.

2. If ZBP is required for RIPK1 recruitment to TRIF complexes, as the authors conclude, then is RIPK1 S166 phosphorylation and degradation upon LPS/TAK1 inhibitor treatment blocked to the same extent as shown for TRIF deficient cells (Figure 1)?

In order to address this question, in the revised manuscript we included blots showing RIP1 total levels and RIP1 S166 phosphorylation. Similar to *Trif*^{-/-}, degradation of RIP1 and S166 phosphorylation is delayed and decreased in the absence of ZBP1.

3. Figure 1A. The authors note “we identified the TLR4 adaptor TRIF as an important mediator of pyroptosis (Figure 1A).”, as their manuscript title also alludes to. It would benefit this take-home conclusion if the authors directly compare, at the same time, LPS/TAK-1 inhibition killing in TRIF vs. GSDMD KO BMDMs, as TRIF can also engage apoptotic caspases, as the authors go on to demonstrate in Figure 1B, and I would be surprised if the death is not a combination of both apoptosis and pyroptosis; which should be clearly resolved for readers and the text adjusted if this is found to be true (as hinted at by the authors own data presented in figure 2C). Moreover, relative to apoptotic caspase cleavage, the cleavage of GSDMD to the 30 kDa pore-forming fragment is not apparent in Figure 1B (that the authors indicate is representative of 3 independent experiments), which would suggest that GSDMD is not even activated.

In the revised version of the manuscript we have included a panel (1A) which demonstrates the relative contribution of CASP8, CASP3/CASP7, GSDMD, RIP3 and RIP1 kinase activity in LPS/5z7 induced cell death. We fully agree that LPS/5z7 induces cell death that is a combination of apoptosis and pyroptosis, and we hope that the addition of the described data in Figure 1, as well as the adjustments made in the text will make this point more clear. Regarding blots for GSDMD in Figures 1 and 2, these blots have been replaced with blots that we feel are more representative of the phenotype for GSDMD cleavage that we typically observe and more in line with our previously published data (*Sarhan et al. PNAS, 2018*).

Replacement blots were run shortly after initial submission using the same lysates but with a different lot of GSDMD antibody.

4. Figure 1A/B. The authors data demonstrate that over-time TRIF-independent killing can take place. Does this reflect autocrine TNF-TNFR1 activation or is the TAK1 inhibitor toxic (inhibitor alone and LPS alone treatment control data is not shown)? It is also unclear how caspase-9 comes to be activated with similar kinetics to caspase-8 in LPS/TAK1 inhibitor treated cells (Figure 1B), as caspase-9 processing is a readout of mitochondrial apoptosis not typically associated with LPS killing.

We agree that over time TRIF-independent killing does occur. This death occurs to a similar extent as that induced by treatment with 5z7 alone (now shown in Figure 1B). We have shown previously that 5z7-induced cell death is dependent on TNF-TNFR1 activation (*Sarhan et. al, PNAS, 2018*). Additionally, in the revised version of the manuscript we show that the remaining cell death that occurs in response to LPS/5z7 treatment in TRIF-deficient macrophages can be largely abrogated by treatment with TNF-neutralizing antibody. Although we do not know the specific mechanism for caspase-9 activation in the context of LPS/5z7 treatment, in the revised manuscript we have confirmed that caspase-9 cleavage is dependent on caspase-8 (Figure S1). Since it has been shown that GSDMD is capable of driving mitochondrial permeabilization, amplifying apoptotic and pyroptotic cell death, we were interested to see if GSDMD was required for CASP9 cleavage in response to LPS/5z7. However, while CASP9 cleavage was dependent on CASP8, it occurred independently of GSDMD (Figure S1). It is likely that caspase-9 activation occurs as a result of mitochondrial dysfunction in response to TAK1 inhibition (Kaiser, *J Biol Chem*, 2013) (Wang JS, *J Biomed*, 2015).

5. Figure 2B/C. As noted above, it is important to define how much TRIF/ZBP1 independent death is TNF driven (e.g. via neutralizing TNF antibodies and/or deletion on top of TRIF KO and ZBP1 KO cells [using appropriate controls]).

As described above, we have used TNF-neutralizing antibodies to analyze the extent that TRIF/ZBP1 independent death is driven by TNF-signaling. Specifically in Figure 4C we have shown that in the absence of TRIF or ZBP1, treatment with TNF-neutralizing antibody does have a modest impact on cell death, indicating that TNF-signaling may amplify cell death in the absence of sufficient TRIF/ZBP1 signaling.

6. The authors write “To further characterize cell death in *Zbp1*^{-/-} BMDMs, we observed that although Annexin V and propidium iodide (PI) staining was delayed in the absence of ZBP1, PI incorporation and weak Annexin V staining occurred nearly simultaneously in both B6 and *Zbp1*^{-/-} BMDMs, consistent with the pyroptotic mechanism of CASP8-mediated cell death (Figure S1F).” It is unclear to me how PI staining can be delayed in the absence of ZBP1 yet occur simultaneously? Do they mean that ZBP1 deficient cells eventually become PI positive? How is this “consistent with the pyroptotic mechanism of casp8-mediated cell death”?

We have removed this data from the revised manuscript since we agree that the statement regarding the mechanism of cell death in ZBP1-deficient BMDMs was confusing, and feel that the data did not significantly contribute to the overall message of our manuscript.

7. Figure 2D. Caspase-8 and caspase-3 cleavage is not altered by *Zbp1* loss, while pyroptotic GSDMD p30 cleavage is not significantly induced by LPS/TAK-inhibitor treatment (as per Figure 1A). Therefore this data does not agree with the authors conclusions that these cleavage events “were delayed by ZBP1 deficiency (Figures 1B, 2D).” and a “pyroptotic cell death mechanism”.

Although ZBP1-deficiency does not fully abrogate cleavage of caspase-8 and caspase-3, the cleavage of caspase-8 and caspase-3 is delayed until 4 hours after treatment with LPS/5z7 in ZBP1-KO BMDMs, compared to 3 hours in B6 macrophages, which altogether is consistent with the delay of cell death in ZBP1-KO BMDMs observed by propidium iodide incorporation (Figure 2B). Similarly, in the revised GSDMD blot (Figure 2E, changes described above), robust cleavage of GSDMD in the absence of ZBP1 occurs around 4 hours after treatment compared to 3 hours in B6. However, we fully agree that even in the absence of ZBP1, cleavage of these caspases occurs and cell death progresses. Therefore, we claim that ZBP1 enhances but is not essential for LPS/5z7 induced cell death.

8. Figure 2E-2G. Decreased IL-1 release upon casp8 deficiency may simply reflect impaired inflammasome priming (e.g. LPS-induced NLRP3 and precursor IL-1 levels), as many studies have reported. Similarly, measuring precursor and bioactive (i.e. cleaved) caspase-1 and IL-1 levels in ZBP1 KOs is required, and showing this is NLRP3 dependent (e.g. genetically or via MCC950 inhibition) is important for the authors conclusions. It is also wrong to conclude that “This decrease in inflammasome-mediated IL-1 β release was likely due to the delayed kinetics of CASP8 activation observed in *Zbp1*^{-/-} BMDMs.” when, in fact, Figure 2D suggests equivalent, or even more, caspase-8 is cleaved in ZBP1 deficient cells.

We agree that decreased IL-1 β release upon CASP8 deficiency can be attributed to impaired inflammasome priming as we have shown previously (Muendlein et. al, Science, 2020). In the revised version of the manuscript we have included blots for pro-IL-1 β , CASP1 and CASP11 to assess the role of ZBP1 in inflammasome priming (Figure 2G). In the revised manuscript we have included data to indicate the requirement for the NLRP3 inflammasome using both LPS/5z7 treated NLRP3-deficient macrophages and treatment of B6, *Trif*^{-/-}, *Tnfr*^{-/-} and *Zbp1*^{-/-} macrophages with MCC950. Additionally, the sentence attributing the decrease in IL-1 β release in the absence of ZBP1 to the delayed kinetics of CASP8 activation has been removed.

9. Figure 3A/F and elsewhere (e.g. figure 4). Essential input and ZBP KO controls are lacking.

Input blots for all immunoprecipitation experiments have been included in the revised manuscript.

10. Figure 4A. As indicated above, the significant cell death observed in TRIF KO cells at later time points might still reflect TNFR1 signaling and this should be tested.

As described above, we have used TNF-neutralizing antibodies to analyze the extent of cell death that can be attributed to TNF-signaling in the absence of TRIF/ZBP1. In Figure 4C we have shown that in the absence of TRIF or ZBP1, treatment with TNF-neutralizing antibody does have a modest impact on cell death, indicating that TNF-signaling may amplify cell death in the absence of sufficient TRIF/ZBP1 signaling.

11. If, as the authors allude ZBP1 is critical for the “TRIFosome” response, do ZBP-1 deficient macrophages, like TRIF deficient cells, show reduced LPS-induced IRF/IFN β responses and inflammatory cytokine production (e.g. see DOI: 10.1126/science.1087262). Is TLR3-TRIF signalling, including TLR3-induced cytokine and killing responses also altered in ZBP-1 deficient cells? Are ZBP deficient animals resistant to endotoxic shock, akin to TRIF KO animals?

In response to this reviewer comment, we have analyzed TNF and IFN β mRNA levels and found that they are decreased in ZBP1 deficient macrophages in response to LPS and LPS/5z7 (Figure S3A). Additionally, we have shown that cell death in response to the TLR3 ligand Poly I:C +5z7 treatment is dependent on ZBP1.

REVIEWER COMMENTS

Reviewer #1 (Remarks to the Author):

As described previously, this manuscript by Muendlein and colleagues identifies a novel ZBP1-containing complex that is required for efficient induction of Casp8-mediated pyroptosis. This finding sheds further light on Casp8-mediated pyroptosis as well as on another role for ZBP1 during cell death and organismal development. Overall, the authors did an excellent job addressing all of the previously listed concerns. All of their additional data further strengthens and supports their previous findings. Overall, this manuscript is greatly improved and the data is very convincing and robust. There were a few minor points to be addressed. The immunoprecipitation of native complexes from primary macrophages is an important advance for the field as well. Several minor (primarily text-based) changes are suggested to improve the manuscript.

1. Fig 4C uses a TNF neutralizing antibody in LPS/5z7-treated B6, *Trif*^{-/-}, *Tnfr*^{-/-}, and *Zbp1*^{-/-} BMDMs to determine the possible role for TNF signaling in this cell death pathway. The authors state that the TNF antibody "decreased death in *Trif*^{-/-} and *Zbp1*^{-/-} BMDMs, indicating that TNFR1-signaling may contribute to cell death in the absence of sufficient TRIF-mediated signaling." Given the lack of statistics on these graphs, it is difficult to determine if this is a statistically significant difference. Also, given the small difference in the deaths, it is difficult to determine if this is a physiologically significant difference. Further, it would have been valuable to have a positive control for TNF signaling in order to determine if the α -TNF is actually working and blocking TNF signaling. However, given the *Tnfr*^{-/-} data, I don't think that any change in 4C is critical. A brief mention of these points in the text would be worthwhile.

2. Related to this - it is surprising given that the authors are looking at the role of *Zbp1* in mediating the intersection between the TNF and TRIF pathways, that they do not cite the work of Peterson et al, JI 2016, which demonstrated that yersinia infection induces this Casp8-dependent death pathway via synergy between TRIF and TNFR signaling. Moreover, that there is a cell-extrinsic TNF signal that synergizes with TRIF to facilitate this death. To the extent that pharmacological inhibition with 5Z7 is supposed to mimic the activity of YopJ, it is important that this prior work be referenced in this context. Additionally, the physiological relevance of this pathway was demonstrated in studies by Peterson et al., JEM 2017, and by Dondelinger et al., Nature Comms, 2019. These studies should be cited in this work as well.

3. Page 7 Line 143: The data suggests that *ZBP1*^{-/-} BMDMs treated with LPS have attenuated S321-phosphorylation on RIPK1. It is undrestandable as to why there might be a reduction in phospho-S166 in the *Zbp1*-knockout, but how does ZBP1 contribute to the negative regulatory phosphorylation at S321? This was not clear in the text. Can the authors please provide a discussion of this with regard to how they think ZBP1 might orchestrate the balance of pro- and anti-death post-translational modifications of RIPK1 within the TRIF-osome?

4. The authors should perform statistics on Fig 4A-D and Fig S3C.

Reviewer #2 (Remarks to the Author):

The authors have somewhat addressed the initial concerns raised. However, several remain:

1. The authors need to indicate molecular weight markers on their blots. This can easily be done in software such as photoshop. Without this it is impossible to judge their accuracy, other than taking

the authors word for it (which is not good scientific practice and is why journals often now stipulate that Mw markers must be included and/or ask for whole-blot scans of the probings).

2. The authors note "We fully agree that LPS/5z7 induces cell death that is a combination of apoptosis and pyroptosis". However, this important sentiment is not reflected in the manuscript title or abstract.

3. Figure 1A. It is strange that the casp3/7 DKO BMDMs die, but plateau at ~60% death within a few hours and do not appear to die further. Is this result consistent across independent experiments?

4. Figure 1B shows that 5z7 treatment alone is toxic. This makes data interpretation of LPS/5z7 treatment across other genotypes/data panels difficult, as these all lack this important internal control. i.e. are observed genotype differences a result of altered resistance/sensitivity to 5z7 toxicity alone, or a result of LPS/5z7 killing? Does 5z7 alone alter ZBP1 and RIPK1 complexes? This represents my major concern with the current interpretation of data and conclusions (e.g. the title states that ZBP1 promotes LPS induced pyroptosis....).

5. Figure 1C. This blot is difficult to interpret because there is a white splotch obscuring the caspase-8 cleavage fragments in the Trif KOs.

6. Figure 2E. There is only a 1hr delay in caspase-8 and GSDMD cleavage between WT and ZBP1 deficient cells. This contrasts the much more significant delay in GSDMD cleavage upon TRIF loss (the caspase-8 blot is obscured by a white blotch so it is difficult to ascertain what is going on here, unfortunately). Therefore, ZBP appears to play a minor role in GSDMD-mediated pyroptosis, and the title/abstract should not overstate the role of ZBP1 in pyroptotic death.

Below is our point-by-point response to the concerns of the reviewers. The reviewers' comments are highlighted in gray followed by our response. In the revised manuscript, all edits are also in gray.

Reviewer #1 (Remarks to the Author):

As described previously, this manuscript by Muendlein and colleagues identifies a novel ZBP1-containing complex that is required for efficient induction of Casp8-mediated pyroptosis. This finding sheds further light on Casp8-mediated pyroptosis as well as on another role for ZBP1 during cell death and organismal development. Overall, the authors did an excellent job addressing all of the previously listed concerns. All of their additional data further strengthens and supports their previous findings. Overall, this manuscript is greatly improved and the data is very convincing and robust. There were a few minor points to be addressed. The immunoprecipitation of native complexes from primary macrophages is an important advance for the field as well. Several minor (primarily text-based) changes are suggested to improve the manuscript.

It is exciting that the reviewer appreciates our effort towards making a convincing and complete story. We are happy to address the remaining suggestions of this reviewer.

1. Fig 4C uses a TNF neutralizing antibody in LPS/5z7-treated B6, *Trif*^{-/-}, *Tnfr*^{-/-}, and *Zbp1*^{-/-} BMDMs to determine the possible role for TNF signaling in this cell death pathway. The authors state that the TNF antibody "decreased death in *Trif*^{-/-} and *Zbp1*^{-/-} BMDMs, indicating that TNFR1-signaling may contribute to cell death in the absence of sufficient TRIF-mediated signaling." Given the lack of statistics on these graphs, it is difficult to determine if this is a statistically significant difference. Also, given the small difference in the deaths, it is difficult to determine if this is a physiologically significant difference. Further, it would have been valuable to have a positive control for TNF signaling in order to determine if the a-TNF is actually working and blocking TNF signaling. However, given the *Tnfr*^{-/-} data, I don't think that any change in 4C is critical. A brief mention of these points in the text would be worthwhile.

In the revised manuscript we have performed statistical analyses on these experiments in which TNF neutralizing antibody is used. Additionally, we have included a panel (Figure 4D) in which TNF neutralizing antibody inhibits 5z7 induced cell death (which we have previously shown to be dependent on TNF (Sarhan et al., 2018)) as a positive control. Finally, we have included a discussion of the potential physiological relevance of this TNF-dependent decrease in cell death in the text.

2. Related to this - it is surprising given that the authors are looking at the role of *Zbp1* in mediating the intersection between the TNF and TRIF pathways, that they do not cite the work of Peterson et al, JI 2016, which demonstrated that yersinia infection induces this Casp8-dependent death pathway via synergy between TRIF and TNFR signaling. Moreover, that there is a cell-extrinsic TNF signal that synergizes with TRIF to facilitate this death. To the extent that pharmacological inhibition with 5Z7 is supposed to mimic the activity of YopJ, it is important that this prior work be referenced in this context. Additionally, the physiological relevance of this pathway was demonstrated in studies by Peterson et al., JEM 2017, and by Dondelinger et al., Nature Comms, 2019. These studies should be cited in this work as well.

We agree that these previous works are highly important to mention in the context of our findings. These studies have been cited in the revised manuscript, refs. 34, 35, 36.

3. Page 7 Line 143: The data suggests that ZBP1^{-/-} BMDMs treated with LPS have attenuated S321-phosphorylation on RIPK1. It is understandable as to why there might be a reduction in phospho-S166 in the Zbp1-knockout, but how does ZBP1 contribute to the negative regulatory phosphorylation at S321? This was not clear in the text. Can the authors please provide a discussion of this with regard to how they think ZBP1 might orchestrate the balance of pro- and anti-death post-translational modifications of RIPK1 within the TRIFosome?

In the current version of the manuscript we have included a discussion of potential ways in which ZBP1 may regulate these RIPK1 modifications within the TRIFosome, page 12, line 8.

4. The authors should perform statistics on Fig 4A-D and Fig S3C.

In the revised version of the manuscript we have performed statistics on these figures.

Reviewer #2 (Remarks to the Author):

The authors have somewhat addressed the initial concerns raised. We would like to stress that we experimentally addressed 11 out of the 12 initial concerns raised by the reviewer, thus putting together a significant amount of requested data. The concern regarding the molecular weight markers was badly overlooked, and I sincerely apologize for that on behalf of all the authors. However, several remain. Below we continue addressing the initial and additional concerns of the reviewer:

1. The authors need to indicate molecular weight markers on their blots. This can easily be done in software such as photoshop. Without this it is impossible to judge their accuracy, other than taking the authors word for it (which is not good scientific practice and is why journals often now stipulate that Mw markers must be included and/or ask for whole-blot scans of the probings).

The molecular weight markers have been added to all blots and the uncropped versions of all western blots with markers are provided in the supplementary documents.

2. The authors note “We fully agree that LPS/5z7 induces cell death that is a combination of apoptosis and pyroptosis”. However, this important sentiment is not reflected in the manuscript title or abstract.

In the revised version of the manuscript, “pyroptosis” has been changed to “cell death”, including in the title and abstract.

3. Figure 1A. It is strange that the casp3/7 DKO BMDMs die, but plateau at ~60% death within a few hours and do not appear to die further. Is this result consistent across independent experiments?

This result is consistent across >15 independent experiments. Furthermore, it is not uncommon for a certain percentage of the population to be protected from cell death due to slight variations in the levels of prosurvival factors from cell to cell, as we have shown recently (Muendlein et al., 2020). It is possible that this is exposed to a greater extent in the absence of CASP3/7.

4. Figure 1B shows that 5z7 treatment alone is toxic. This makes data interpretation of LPS/5z7 treatment across other genotypes/data panels difficult, as these all lack this important internal control. i.e. are observed genotype differences a result of altered resistance/sensitivity to 5z7 toxicity alone, or a result of LPS/5z7 killing? Does 5z7 alone alter ZBP1 and RIPK1 complexes? This represents my major concern with the current interpretation of data and conclusions (e.g. the title states that ZBP1 promotes LPS induced pyroptosis....).

Indeed, treatment of BMDMs with 5z7 alone results in cytotoxicity driven mostly via TNF-dependent necroptosis as we reported earlier using RIPK inhibitors (Sarhan et al., 2018). This finding is further supported by the fact that MLKL-deficient BMDMs are resistant to 5z7-induced cytotoxicity as compared to wild type BMDMs, Figure 1B below. In contrast, MLKL has no role in LPS/5z7-induced cell death, Figure 1A below, thus defining LPS/5z7-induced cell death as distinct from that induced by 5z7 alone. Additionally, ZBP1 does not affect 5z7-induced necroptosis, Figure 1D, as compared to LPS/5z7 cell death, Figure 1C, demonstrating that the role of ZBP1 is specific to LPS/5z7-induced cell death. Finally, TRIF-deficiency attenuates LPS/5z7 induced cell death, which supports the mechanism in which both ZBP1 and TRIF specifically contribute to the same mechanism of CASP8-mediated cell death. We hope that this data addresses the reviewer's concerns regarding the role of ZBP1 in LPS/5z7-mediated cell death as opposed to 5z7 alone. We are currently working on an additional story characterizing the mechanism of 5z7 induced cell death, and the role of TRIF and TNFR1 in this process. Therefore, while we are happy to share this information with the reviewer's, we would prefer to avoid including this data in the current manuscript if the reviewers are agreeable.

5. Figure 1C. This blot is difficult to interpret because there is a white splotch obscuring the caspase-8 cleavage fragments in the Trif KO's.

We agree that the white splotch on this blot made interpretation of the kinetics of caspase-8 cleavage difficult and we apologize. We have rerun this western blot and hope the data is easier to interpret.

6. Figure 2E. There is only a 1hr delay in caspase-8 and GSDMD cleavage between WT and ZBP1 deficient cells. This contrasts the much more significant delay in GSDMD cleavage upon TRIF loss (the caspase-8 blot is obscured by a white blotch so it is difficult to ascertain what is going on here, unfortunately). Therefore, ZBP appears to play a minor role in GSDMD-mediated pyroptosis, and the title/abstract should not overstate the role of ZBP1 in pyroptotic death.

As previously noted, we switched all mentions of "pyroptosis" to cell death in the title and throughout the text. Furthermore, we specifically state on page 7, line 1, of the revised manuscript that ZBP1 is involved in the regulation of both apoptotic and pyroptotic cell death pathways" ... Regarding the role of ZBP1, the deletion of ZBP1 rescues almost 50% of cells that otherwise die in response to LPS/5z7, which in our opinion constitutes a significant rather than minor role of ZBP1 in cell death. We have ensured that nowhere in the paper do we define the effect of ZBP1 as major, yet we cannot conclude that this is a minor effect because a 50% rescue is significant. The first version of the title had "...ZBP1 initiates..." which indicates sensing, which is not the case for this particular mechanism of cell death. Therefore, we changed to "promotes" in order to avoid implying that ZBP1 uses its sensor features. We hope that in the revised version of the manuscript the reviewer does not find the role of ZBP1 to be overstated.

Sarhan, J., et al. (2018). Caspase-8 induces cleavage of gasdermin D to elicit pyroptosis during Yersinia infection. *Proc Natl Acad Sci U S A* 115, E10888-E10897.

Muendlein, H. I. et al. cFLIPL Protects Macrophages From LPS Induced Pyroptosis via Inhibition of Complex II Formation. *Science* (80-.). 1379–1384 (2020).

REVIEWERS' COMMENTS

Reviewer #1 (Remarks to the Author):

The authors have addressed all of the remaining comments. This manuscript will provide important new insight into regulation of cell death in the setting of TAK1 disruption.

Reviewer #2 (Remarks to the Author):

I would like to thank the authors for clarifying and addressing the concerns raised; which they have now done so. I can understand that the authors wish to delineate the mechanism of 5z7 killing in a separate story. However, for clarity of the current story it would be of benefit to show some of the data presented in the reviewers response figure 1D - documenting that 5z7 killing by itself is comparable between WT and ZBP1 deficient cells; it is important to show readers that the difference between WT and ZBP1 responses to LPS/5z7 treatment are not simply the result of 5z7 toxicity.

Reviewer #1 (Remarks to the Author):

The authors have addressed all of the remaining comments. This manuscript will provide important new insight into regulation of cell death in the setting of TAK1 disruption.

We are happy to hear that we have addressed all comments, and that the reviewer feels our findings provide important new insight into the regulation of cell death in this setting.

Reviewer #2 (Remarks to the Author):

I would like to thank the authors for clarifying and addressing the concerns raised; which they have now done so. I can understand that the authors wish to delineate the mechanism of 5z7 killing in a separate story. However, for clarity of the current story it would be of benefit to show some of the data presented in the reviewers response figure 1D - documenting that 5z7 killing by itself is comparable between WT and ZBP1 deficient cells; it is important to show readers that the difference between WT and ZBP1 responses to LPS/5z7 treatment are not simply the result of 5z7 toxicity.

We are happy to hear that the reviewer feels we have sufficiently clarified and addressed all concerns. We understand that showing that 5z7-induced cell death is not dependent on ZBP1 adds clarity to our story and rules out a role for ZBP1 in 5z7 toxicity. To that end, we have included cell death curves from B6 and ZBP1-deficient macrophages stimulated with 5z7 alone in the revised version of the manuscript (Figure S2F).

Alexander Poltorak, Ph.D.
Professor and Chair
Department of Immunology
Tufts University School of Medicine
Boston, MA 02111